# Bridging Crypto with ML-based Solvers: the SAT Formulation and Benchmarks

**Xinhao Zheng**[*], **Xinhao Song**[*], **Bolin Qiu, Yang Li, Zhongteng Gui, Junchi Yan**[‡]
School of Computer Science & School of AI, Shanghai Jiao Tong University
Shanghai Innovation Institute
{void_zxh,sxh001,boring123,yanglily,dragon-archer,yanjunchi}@sjtu.edu.cn
https://github.com/void-zxh/SAT4CryptoBench

## Abstract

The Boolean Satisfiability Problem (SAT) plays a crucial role in cryptanalysis, enabling tasks like key recovery and distinguisher construction. Conflict-Driven Clause Learning (CDCL) has emerged as the dominant paradigm in modern SAT solving, and machine learning has been increasingly integrated with CDCL-based SAT solvers to tackle complex cryptographic problems. However, the lack of a unified evaluation framework, inconsistent input formats, and varying modeling approaches hinder fair comparison. Besides, cryptographic SAT instances also differ structurally from standard SAT problems, and the absence of standardized datasets further complicates evaluation. To address these issues, we introduce SAT4CryptoBench, the first comprehensive benchmark for assessing machine learning–based solvers in cryptanalysis. SAT4CryptoBench provides diverse SAT datasets in both Arithmetic Normal Form (ANF) and Conjunctive Normal Form (CNF), spanning various algorithms, rounds, and key sizes. Our framework evaluates three levels of machine learning integration: standalone distinguishers for instance classification, heuristic enhancement for guiding solving strategies, and hyperparameter optimization for adapting to specific problem distributions. Experiments demonstrate that ANF-based networks consistently achieve superior performance over CNF-based networks in learning cryptographic features. Nonetheless, current ML techniques struggle to generalize across algorithms and instance sizes, with computational overhead potentially offsetting benefits on simpler cases. Despite this, ML-driven optimization strategies notably improve solver efficiency on cryptographic SAT instances. Besides, we propose BASIN, a bitwise solver taking plaintext-ciphertext bitstrings as inputs. Crucially, its superior performance on high-round problems highlights the importance of input modeling and the advantage of direct input representations for complex cryptographic structures.

## 1 Introduction

The Boolean Satisfiability Problem (SAT) is a fundamental problem in computer science, offering a powerful approach to express and analyze cryptographic problems. With the advent of Conflict-Driven Clause Learning (CDCL), modern SAT solvers have demonstrated remarkable efficiency across diverse application domains. It provides a natural bridge between cryptanalysis and advanced solving techniques. By representing cryptographic challenges as SAT instances, researchers can leverage state-of-the-art SAT solvers to tackle complex cryptographic problems[1, 2, 3]. This connection enables cryptanalysis to benefit from efficient SAT-solving strategies and makes cryptographic challenges more accessible to researchers from the algorithmic and machine-learning communities [4, 5, 6].

---

[*] denotes equal contribution. [‡]Correspondence author. Work was partly supported by NSFC (92370201).

However, benchmarking machine learning (ML) based SAT solvers on cryptographic instances remains challenging. Existing solvers vary widely in input formats, modeling paradigms, and ML component integrations, complicating direct comparison. Additionally, cryptographic SAT instances often exhibit domain-specific structural properties, such as high symmetry and algebraic dependencies, which distinguish them fundamentally from existing general SAT benchmarks. Furthermore, the lack of standardized datasets and evaluations across different input formats hinders reproducible and fair assessment of solver performance. These challenges highlight the pressing need for a unified benchmark specifically designed for ML-based SAT solvers in cryptographic applications.

To address these challenges, we propose **SATCryptoBench**, a comprehensive benchmark for evaluating ML-based SAT solvers on a range of cryptographic problems. SATCryptoBench includes a large collection of SAT datasets derived from various cryptographic algorithms, and generates datasets designed to contain instances of multiple representations: Arithmetic Normal Form (ANF), Conjunctive Normal Form (CNF), and the original plaintext-ciphertext bitstrings. These representations serve two main purposes: (1) to enable systematic studies on how problem formulation affects solver performance—for example, ANF is often more natural for expressing cryptographic structures; and (2) to improve accessibility for the SAT solving community, where CNF remains the standard despite its limitations in modeling cryptographic operations. Through this design, SATCryptoBench bridges the gap between traditional cryptanalysis and modern SAT-solving techniques.

Then, in order to systematically analyse the performance of different ML-based solvers on SAT instances of the encryption problem, SATCryptoBench classifies the solvers into three levels according to the ML integration in the solving process:

- **Standalone Distinguisher**: Solvers that independently classify SAT instances (satisfiable or unsatisfiable) or identify cryptographic properties without directly solving the SAT problem.
- **Heuristic Enhancement**: Solvers that incorporate machine learning to guide traditional heuristics.
- **Hyperparameter Optimization**: Solvers that utilize machine learning to optimize hyperparameters, enabling heuristics to better adapt to specific cryptographic problem distributions, without requiring knowledge of the solver's internal structure.

Experiments are conducted to evaluate ML-based SAT solvers across the above three levels. For Standalone Distinguisher, we assess 8 models on 6 cryptographic datasets, covering a range of rounds and key sizes. For Heuristic Enhancement, we evaluate 10 ML-based solvers alongside their original counterparts across 11 datasets, encompassing diverse cryptographic algorithms and instance scales. For Hyperparameter Optimization, we compare the performance of three popular traditional solvers under various hyperparameter optimization strategies on large-instance datasets. Furthermore, we propose a bitwise solver, BASIN, which takes plaintext-ciphertext bitstrings as input and performs a comparative evaluation of different input formulations for ML-based solvers. **Our empirical analysis highlights the following key findings:**

1. ANF-based networks excel at capturing cryptographic features, achieving significantly higher prediction accuracies compared to CNF-based approaches across our benchmark datasets.
2. ML models struggle to generalize across different cryptographic algorithms and instance sizes, and the integration of ML components introduces computational overhead that can offset performance improvements, particularly for less complex instances.
3. Different machine learning-based optimization strategies, despite their varying approaches, can all significantly improve solver performance on cryptographic instances
4. The bitwise solver, BASIN, outperforms its ANF- and CNF-based counterparts on high-round cryptographic instances, suggesting that direct input representations, such as plaintext-ciphertext bitstrings, can more effectively capture the underlying cryptographic structures in complex settings.

## 2   Related Work

**Learning-assisted heuristic solvers.** Machine learning has shown significant potential in augmenting SAT solvers by integrating data-driven heuristics [4, 5, 6]. Learning-assisted methods [7, 8] enhance traditional CDCL-based solvers by embedding learned components into specific solving stages. Among these, the most extensively studied area is branching heuristics [9, 10], where neural models guide variable selection decisions. Beyond branching, machine learning has also been applied to optimize variable initialization [11, 12], clause deletion [13], restart policies [14], and glue clause

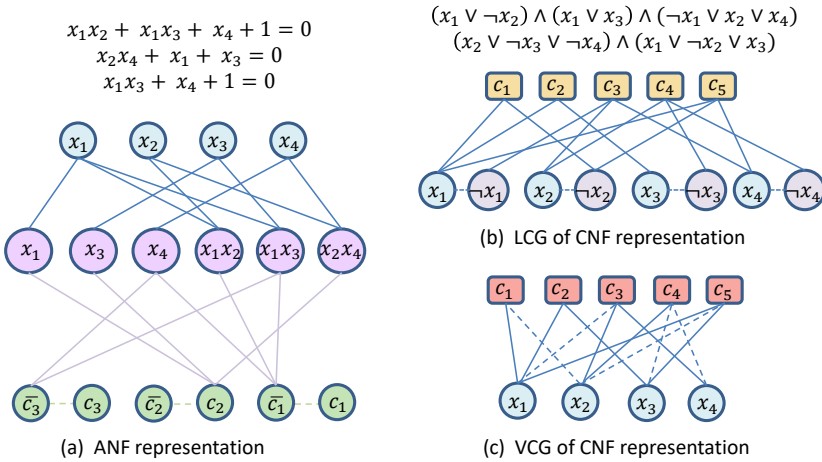

$$x_1x_2 + x_1x_3 + x_4 + 1 = 0$$
$$x_2x_4 + x_1 + x_3 = 0$$
$$x_1x_3 + x_4 + 1 = 0$$

(a) ANF representation

$$(x_1 \vee \neg x_2) \wedge (x_1 \vee x_3) \wedge (\neg x_1 \vee x_2 \vee x_4)$$
$$(x_2 \vee \neg x_3 \vee \neg x_4) \wedge (x_1 \vee \neg x_2 \vee x_3)$$

(b) LCG of CNF representation

(c) VCG of CNF representation

Figure 1: Graph representation. (a) ANF graph representation of the SAT instances. (b) Literal-Clause Graph (LCG) of CNF representation. (c) Variable-Clause Graph (VCG) of CNF representation.

prediction [15, 16]. These approaches strategically leverage learned components while preserving the robustness of classical solvers. However, most of these methods have not been evaluated on large-scale cryptographic SAT datasets, and there remains a critical gap: the lack of a unified dataset and evaluation framework for cryptographic SAT problems, which hinders systematic comparison of learning-assisted methods in this important application domain. Our SAT4CryptoBench aims to fill this gap by evaluating various learning-assisted solvers under a unified benchmark.

**End-to-end neural solvers.** Another major direction of applying machine learning to SAT solving is the development of end-to-end neural solvers, which regard SAT solving as a prediction problem, bypassing traditional solving pipelines. Early attempts used recursive neural networks to process CNF formulas as sequences [17], which later evolved into more powerful graph neural network (GNN)-based approaches capable of directly capturing formula structures through graph representations. This line of work mainly focuses on satisfiability prediction [18, 19, 20, 21, 22], eliminating the need for hand-crafted features used in earlier methods like SATzilla [23]. Additionally, to address data scarcity, the community has introduced auxiliary tasks such as generating pseudo-industrial instances [24, 25, 26]. Besides, Li et al. [27] propose a benchmark for evaluating GNN-based SAT solving. However, existing works mainly target CNF-based networks and general SAT problems, while cryptographic SAT instances often exhibit unique structures that are hard to efficiently represent in CNF, leading to a scale explosion and loss of semantics. Recently, Zheng et al. [2] propose an ANF-based end-to-end solver specifically designed for cryptographic SAT problems, highlighting the growing interest in this field. These developments highlight the necessity and challenge of establishing a systematic evaluation framework that supports both CNF-based and ANF-based end-to-end solvers for cryptographic problems. To this end, our SAT4CryptoBench provides unified datasets in both formats and enables comprehensive evaluation of neural solvers on cryptographic SAT instances.

**SAT for Cryptography.** SAT solving has become important in automated cryptanalysis. Nejati et al. [1] introduce CDCL (Crypto), a framework that enhances CDCL solvers with domain-specific cryptographic knowledge through programmatic callbacks. Similarly, Lafitte et al. [28] developed CryptoSAT, an open-source framework that automates SAT-based analysis for cryptographic primitives, simplifying the conversion of cryptographic algorithms into CNF formulas. Other SAT-based cryptanalysis works include exploring differential trails in ARX ciphers [29] and developing automated search techniques for ciphers with S-boxes to achieve more accurate differential trails [30]. Additionally, an SAT-based automated search toolkit [3] has been applied to SHA-3. However, existing studies often focus on case studies or address only small components of specific cryptographic problems, which limits consistent and meaningful cross-method evaluation. A unified framework for systematically evaluating these approaches in cryptographic problem settings remains absent.

To address the challenges of evaluating diverse SAT-solving approaches in cryptographic problems, particularly with the emergence of machine learning-based methods, we propose a comprehensive

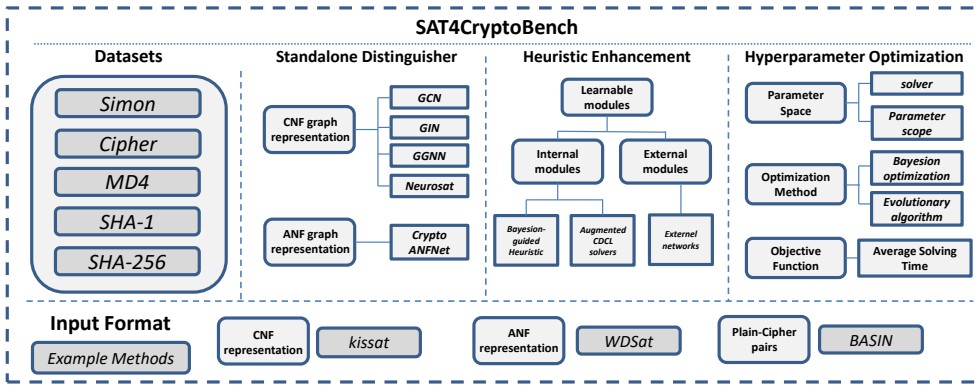

Figure 2: Overview of SAT4CryptoBench.

evaluation framework that standardizes the assessment of SAT solvers across different input formats and problem structures specific to cryptographic applications.

## 3 Preliminaries

**SAT problem & Cryptographic operation.** The SAT problem plays an important role in computational complexity and cryptographic analysis. Cryptographic operations, such as those used in block ciphers, hash functions, and public-key systems, are typically composed of Boolean functions that can be expressed in terms of satisfiability constraints. Specifically, the algebraic structure of cryptographic operations, such as XOR, AND, and modular addition, makes them suitable for translation into satisfiability problems represented in forms like CNF or ANF. Please refer to Appendix B.1 for the detailed representations of common cryptographic operations. By reducing cryptographic problems to SAT instances, researchers can leverage SAT solvers to tackle complex problems like key recovery, collision finding, and differential analysis.

**ANF & CNF graph representation.** Graph representations of SAT problems, both in ANF and CNF forms, provide powerful tools for visualization and analysis. In CNF, an SAT problem is expressed as a conjunction of clauses, with each clause being a disjunction of literals. This naturally maps to a bipartite graph, where one set of nodes represents variables and the other represents clauses. Connections between these nodes indicate the presence of variables (or their negations) in the clauses. This structure facilitates the use of graph-based algorithms for SAT solving and optimization. Figures 1 (b) and (c) show two common graph representations of the CNF formula.

In contrast, the ANF representation, consisting of polynomial equations over GF(2), leads to a different graph structure. In this case, according to [2], variables and monomials in the equations are represented as nodes, with edges capturing the relationships defined by the equations. This graph emphasizes higher-order interactions between variables, making it especially suitable for analyzing cryptographic systems that involve multivariate polynomial equations. For instance, modular addition in encryption schemes introduces quadratic terms, creating second-order dependencies in the ANF graph. Figure 1 (a) illustrates the graph representation of the ANF formula. By combining these graph representations, researchers can gain insights into the structure of cryptographic instances, optimize SAT solver strategies, and develop novel algorithms for specific cryptographic problems.

## 4 Benchmark

The goal of SAT4CryptoBench is to provide a unified framework for evaluating ML-based SAT solvers on cryptographic problems. In this section, based on the ML integration in the solving process, we categorize these solvers into three levels and discuss the details of solvers at each level. Furthermore, we systematically examine how different input representations, such as CNF, ANF, and plaintext-ciphertext bitstrings, affect the solver's performance. The overview is shown in Figure 2.

## 4.1 Standalone Distinguisher

The Standalone Distinguisher represents a data-driven approach that completely replaces traditional SAT-solving algorithms with machine learning methods. These distinguishers use ML models to directly predict the satisfiability of SAT instances or identify cryptographic properties from the input without utilizing any conventional SAT solvers. This paradigm offers a fundamentally different approach, where the entire solving process is replaced by a learned model, enabling rapid assessment of SAT instances, which is particularly valuable for cryptographic analysis.

To evaluate the standalone distinguishers, we construct synthetic datasets based on the Simon block cipher. Each dataset is provided in both ANF and CNF formats, while also retaining the original plaintext-ciphertext bitstrings, offering multiple representations of the same cryptographic problems.

We evaluate two categories of neural network architectures based on their input representations. **For the CNF-based neural network**, the message-passing process is designed to iteratively refine the embeddings of nodes (literals and clauses) by aggregating information from neighboring nodes. Initially, the embeddings of literals and clauses are randomly initialized, denoted by $h_l$ and $h_c$, respectively. During the message-passing iterations, each clause node updates its embedding by receiving messages from connected literal nodes. Taking the LCG representation of a CNF instance as an example, at the $k$-th iteration of message passing, these hidden representations are updated as:

$$
\begin{aligned}
h_c^{(k)} &= \text{UPD}\left(\text{AGG}_{l \in \mathcal{N}(c)}\left(\left\{\text{MLP}\left(h_l^{(k-1)}\right)\right\}\right), h_c^{(k-1)}\right) \\
h_l^{(k)} &= \text{UPD}\left(\text{AGG}_{c \in \mathcal{N}(l)}\left(\left\{\text{MLP}\left(h_c^{(k-1)}\right)\right\}\right), h_{\neg l}^{(k-1)}, h_l^{(k-1)}\right)
\end{aligned}
\tag{1}
$$

where $\mathcal{N}()$ denotes the set of the neighbor nodes. **MLP** denotes the multi-layer perception, **UPD**() represents the update function, and **AGG**() represents the aggregation function.

**For the ANF-based neural network**, due to their tripartite graph representation, the message-passing process is different from CNF-based networks. First, each clause node updates its embedding by receiving messages from connected monomial nodes and its complementary clause:

$$
\begin{aligned}
L_{l2c}^{(t)} &= L_{l2l}(M_{l2l}^T L^{(t)}) \\
[C_{m,\text{pos}}^{(t)}, C_{m,\text{neg}}^{(t)}] &= M_{l2c}^T L_{\text{msg}}(L_{l2c}^{(t)}) \\
(C_{\text{pos}}^{(t+1)}, C_{h,\text{pos}}^{(t+1)}) &\leftarrow C_{u,\text{pos}}([C_{h,\text{pos}}^{(t)}, C_{\text{neg}}^{(t)}, C_{m,\text{pos}}^{(t)}]) \\
(C_{\text{pos}}^{(t+1)}, C_{h,\text{neg}}^{(t+1)}) &\leftarrow C_{u,\text{neg}}([C_{h,\text{neg}}^{(t)}, C_{\text{pos}}^{(t)}, C_{m,\text{neg}}^{(t)}])
\end{aligned}
\tag{2}
$$

where $M_{l2l}$ is the sparse adjacency matrix defining connections between literals and monomials, and $M_{l2c}$ is the sparse adjacency matrix defining connections between literals and clauses. $L_{l2l}, L_{\text{msg}}$ are **MLPs** that process incoming messages and $C_{u,\text{pos}}, C_{u,\text{neg}}$ are update functions. Next, each literal updates its embedding by receiving messages from associated clauses.

$$
L_{c2l}^{(t)} = M_{l2c} C_{\text{msg}}([C_{\text{pos}}^{(t)}, C_{\text{neg}}^{(t)}])
\tag{3}
$$

$$
L_m^{(t)} = M_{l2l} L_{l2m}(L_{c2l}^{(t)})
\tag{4}
$$

$$
(L^{(t+1)}, L_h^{(t+1)}) \leftarrow L_u([L_h^{(t)}, L_m^{(t)}])
\tag{5}
$$

where $L_{l2m}, C_{\text{msg}}$ are **MLPs**, and $L_u$ denotes the update function for the literals' embeddings.

## 4.2 Heuristic Enhancement

SAT4CryptoBench provides a systematic framework for evaluating CDCL SAT solvers that incorporate machine learning components to enhance traditional heuristic decision-making processes. Our framework categorizes these hybrid approaches into two distinct integration patterns:

- **Learnable Internal Modules:** Solvers that incorporate learning components directly into their core framework to guide heuristic decisions during the solving process without pre-training.
- **Neural Network External Modules:** Solvers that utilize separate neural networks trained before the solving process, to enhance specific components of the solving process.

**Heuristic enhancement with learnable internal modules.** This category includes ML-based solvers that integrate learning-based components directly into their algorithmic framework. These solvers enhance their heuristic decisions through embedded learning mechanisms, which dynamically optimize various aspects of the solving process, such as variable selection, branching strategies, clause learning, and restart policies. By incorporating learning mechanisms as fundamental elements of the solving process, these solvers typically do not require pre-training but instead learn to adjust relevant parameters during the solving process, thereby continuously adapting their behavior to the unique characteristics of cryptographic problems. They also leverage domain-specific knowledge to guide their decision-making, improving the efficiency of solving complex SAT instances.

**Heuristic enhancement with learnable external modules.** This category includes solvers that leverage external neural network architectures to improve the solving process. Rather than incorporating learning components directly into the core algorithm, these solvers use neural networks as independent modules that interact with the solver at key decision-making points. Pre-trained to recognize complex patterns in cryptographic instances under specific distributions, these networks can uncover structural features that traditional heuristics may overlook. The learned features enable better branching and clause management, thereby improving overall solving efficiency.

**Input Representation Impact and the BASIN Solver** Most existing SAT solvers rely on CNF as the standard input format. However, given the algebraic and bit-level structure of cryptographic SAT instances, alternative representations—such as ANF or direct plaintext-ciphertext bitstring inputs—may better capture their underlying semantics. To explore this, we examine how different input formats affect solver effectiveness in cryptographic settings.

To this end, we propose **BASIN**, Bitwise Arithmetic Solver with Initialization by Neural network, a bit-level solver architecture specifically designed for cryptographic SAT problems. BASIN belongs to the method of heuristic enhancement, as it combines traditional enumeration with neural-guided initialization to improve solving efficiency. It operates directly on plaintext-ciphertext bitstrings and performs key recovery via guided bitwise search. To enhance initialization, it integrates an improved version of CryptoANFNet, which processes ANF representations derived from input pairs and predicts a likely key assignment. This prediction is used as the initial guess for enumeration, enabling efficient search over the key space.

### 4.3 Hyperparameter Optimization

SAT4CryptoBench offers a comprehensive framework for hyperparameter optimization on cryptographic datasets, aimed at systematically adjusting the hyperparameters of traditional SAT solvers to improve their performance. These methods use machine learning-based optimization algorithms to identify optimal parameter combinations that align with different cryptographic algorithms, fine-tuning solvers' hyperparameters for diverse cryptographic dataset distributions. The framework enables systematic evaluation of hyperparameter optimization methods within broader comparative studies of learning-based SAT solvers. Specifically, the framework comprises three key elements:

- **Parameter Space:** This element defines the optimization parameter space for the solvers, specifying the parameters to be optimized and their respective ranges.
- **Optimization Methods:** This element involves the hyperparameter optimization technique selection. The framework supports various methods, including black-box Bayesian optimization, random search, and evolutionary algorithms, to identify the optimal hyperparameter configurations.
- **Objective Function:** The optimization objective function quantifies the performance of the solver in addressing specific cryptographic SAT problems. Here, we select solving time as the objective function, allowing for a clear assessment of the solver's efficiency.

## 5 Experiment and Discussion

In this section, we conducted extensive experiments across multiple cryptographic algorithms and solver configurations to thoroughly assess the effectiveness of different approaches, as well as the impact of input representations. Detailed experimental settings, including dataset generation and evaluation metrics, are provided in Appendix A.

Table 1: Experimental Results on Simon Block Cipher

| Method | 3 Rounds | | | 6 Rounds | | |
|---|---|---|---|---|---|---|
| | 8-bit | 16-bit | 32-bit | 8-bit | 16-bit | 32-bit |
| GCN [32] (LCG) | 50.5% | 50.0% | 50.0% | 50.0% | 50.0% | 50.0% |
| GCN (VCG) | 52.0% | 52.0% | 59.0% | 55.5% | 50.3% | 51.3% |
| GIN [33] (LCG) | 50.0% | 50.0% | 50.0% | 50.0% | 50.0% | 50.0% |
| GIN (VCG) | 50.0% | 53.0% | 58.0% | 55.8% | 51.5% | 51.0% |
| GGNN [34] (LCG) | 50.0% | 51.0% | 50.0% | 52.3% | 50.3% | 50.0% |
| GGNN (VCG) | 50.0% | 53.5% | 58.0% | 54.8% | 50.8% | 50.0% |
| Neurosat [18] (LCG) | 50.0% | 50.5% | 50.0% | 50.3% | 50.0% | 50.0% |
| CryptoANFNet [2] | **100%** | **99.5%** | **99.5%** | **70.0%** | **69.0%** | **67.3%** |

## 5.1 Datasets

We construct five types of datasets based on different cryptographic problems with both CNF and ANF representations. CNF instances for MD4, SHA-1, and SHA-256 are generated using the SAT encoding toolkit [31], and we generate datasets in both CNF and ANF representations for the Simon cipher to offer complementary views. Additionally, we provide CNF datasets of a widely used but anonymized cryptographic scheme, referred to as Cipher[1], which is derived from real encrypted data using a confidential, widely used block cipher, providing realistic SAT instances. To support fine-grained analysis, we also define multiple difficulty levels by varying encryption rounds and block sizes. The detailed construction process of the datasets is provided in Appendix B.2.

## 5.2 Evaluation of Standalone Distinguisher

We conducted experiments to assess the performance of various distinguisher methods on cryptographic SAT instances. The evaluation included four popular CNF-based neural networks: GCN, GIN, GGNN, and Neurosat, along with a recently proposed ANF-based network, CryptoANFNet [2].

Specifically, we present a comparative analysis of the methods' performance in distinguishing between SAT and UNSAT instance pairs across various Simon configurations. The datasets were derived from the Simon block cipher, configured with encryption rounds of 3 and 6, and block lengths of 8, 16, and 32 bits, resulting in six distinct dataset configurations. Each SAT-UNSAT instance pair in ANF corresponds to a pair in CNF, and both are derived from the same plaintext-ciphertext pair. These instance pairs were carefully constructed to ensure a representative sampling of the problem space.

We set the network settings based on [2, 27]. For the CNF-based networks, GCN, GIN, and GGNN were evaluated using both Literal-Clause Graph and Variable-Clause Graph representations, while Neurosat was evaluated using only the Literal-Clause Graph representation. For the ANF-based network, CryptoANFNet was evaluated using SAT instances in ANF. This evaluation framework offers comprehensive insights into the impact of graph representation on model performance.

From the experimental results in Table 1, we conclude that **the utilization of ANF-based networks for learning features of cryptographic SAT instances demonstrates significant advantages compared to CNF-based networks**. As evidenced by the results in Table 1, across all datasets, different CNF-based models on two CNF-based representations show no significant impact on outcomes. Under CNF-based representations, models are barely able to learn features from cryptographic SAT datasets. In contrast, with ANF-based representations, ANF-based models achieve prediction accuracies substantially higher than their CNF-based counterparts, even reaching prediction accuracies approaching 1 for Simon datasets with 3 rounds. These findings provide compelling evidence supporting our conclusion and further details about CNF representation bottleneck are in Appendix C

## 5.3 Evaluation of Heuristic Enhancement

We conducted experiments to evaluate ML-based heuristic enhancement methods across two categories. The first category includes solvers with internal learnable modules, including Maple-Painless [35], MapleSat [36], MapleSat-Crypto [1], and BMM-enhanced variants [37] (Glucose-

---

[1]The exact name is undisclosed for confidentiality reasons.

Table 2: Average solving time (in seconds) of different SAT solvers with ML-based heuristic enhanced modules and traditional solvers on cryptographic instances. * means the original solver in its publicly available official implementation. Here, we show the 95% confidence interval.

| Method | Cipher 8 | Cipher 9 | Cipher 10 | Cipher 11 | Cipher 12 | MD4 | Simon 10-32-64 | Simon 11-32-64 | Simon 12-32-64 | SHA-1 | SHA 256 |
|---|---|---|---|---|---|---|---|---|---|---|---|
| MaplePainless | 1.07 ± 0.01 | 2.97 ± 0.27 | 21.21 ± 2.21 | 69.40 ± 7.19 | 390.22 ± 46.50 | 1.430 ± 0.035 | 22.00 ± 2.20 | 77.10 ± 7.49 | 445.31 ± 50.86 | 53.89 ± 7.05 | 153.32 ± 23.17 |
| MapleSat | 0.27 ± 0.03 | 1.02 ± 0.12 | 12.94 ± 1.29 | 63.12 ± 9.06 | 381.22 ± 15.94 | 1.435 ± 0.013 | 26.46 ± 5.13 | 73.85 ± 10.2 | 440.02 ± 18.77 | 75.92 ± 13.07 | **111.52** ± 14.41 |
| MapleSat-BMM | 0.38 ± 0.04 | 1.16 ± 0.13 | 12.17 ± 1.51 | 51.29 ± 5.44 | 390.84 ± 45.89 | 0.084 ± 0.001 | 14.83 ± 1.83 | **41.57** ± 5.05 | 460.57 ± 55.87 | 41.75 ± 4.86 | 120.78 ± 18.31 |
| MapleSat-Crypto | 0.34 ± 0.02 | 1.54 ± 0.10 | 14.25 ± 1.07 | **48.36** ± 4.73 | 443.11 ± 35.67 | 0.271 ± 0.018 | 15.98 ± 3.77 | 58.34 ± 5.35 | 717.31 ± 64.83 | 43.21 ± 4.55 | 124.27 ± 16.24 |
| Glucose-BMM | 0.21 ± 0.02 | 1.28 ± 0.12 | 24.56 ± 5.80 | 98.74 ± 14.31 | 1050.95 ± 100.79 | 0.730 ± 0.070 | 27.33 ± 6.32 | 90.46 ± 14.08 | 1388.51 ± 180.96 | 80.96 ± 14.33 | 5764.04 ± 769.49 |
| Glucose* | 0.22 ± 0.02 | 1.36 ± 0.16 | 14.34 ± 1.55 | 70.93 ± 10.91 | 914.88 ± 100.60 | 0.084 ± 0.001 | 14.52 ± 2.02 | 79.15 ± 12.66 | 1434.34 ± 190.33 | 53.25 ± 8.71 | 2932.32 ± 421.31 |
| MapleCOMSPS-BMM | 0.28 ± 0.03 | 1.87 ± 0.26 | 35.32 ± 4.10 | 126.20 ± 15.32 | 1002.57 ± 142.45 | 0.095 ± 0.001 | 32.33 ± 4.20 | 110.60 ± 13.91 | 1248.37 ± 213.52 | 146.34 ± 20.30 | 336.59 ± 47.52 |
| MapleCOMSPS* | 0.22 ± 0.03 | 1.97 ± 0.23 | 29.85 ± 3.65 | 105.94 ± 12.35 | 1061.95 ± 130.22 | 0.102 ± 0.001 | 28.77 ± 3.42 | 125.78 ± 13.28 | 1467.97 ± 220.32 | 194.67 ± 38.04 | 377.19 ± 48.88 |
| MapleLCMDist-BMM | 0.73 ± 0.24 | 3.72 ± 0.55 | 28.66 ± 2.65 | 75.55 ± 7.54 | **368.26** ± 45.72 | 0.084 ± 0.001 | 24.10 ± 2.70 | 63.65 ± 7.44 | **382.07** ± 43.16 | 51.76 ± 5.12 | 226.01 ± 36.77 |
| MapleLCMDist* | **0.11** ± 0.01 | 1.10 ± 0.12 | 21.49 ± 2.48 | 67.88 ± 6.33 | 382.92 ± 40.66 | 0.089 ± 0.001 | 24.41 ± 2.87 | 69.08 ± 6.52 | 389.66 ± 44.37 | **38.40** ± 5.90 | 152.86 ± 21.42 |
| Kissat (NeuroBack) | 10.77 ± 1.59 | 27.48 ± 3.22 | 396.57 ± 63.75 | 1536.74 ± 238.28 | 2184.22 ± 271.85 | 13.539 ± 0.814 | 433.77 ± 61.78 | 1728.98 ± 263.36 | 3783.78 ± 270.93 | 683.39 ± 109.48 | 732.93 ± 65.59 |
| Kissat* | 0.12 ± 0.02 | **0.63** ± 0.07 | **12.14** ± 1.34 | 63.25 ± 4.56 | 974.35 ± 66.02 | **0.078** ± 0.001 | **11.28** ± 1.08 | 73.24 ± 5.10 | 818.92 ± 55.93 | 263.89 ± 14.56 | 293.78 ± 15.21 |
| Neuro-Cadical | 0.95 ± 0.02 | 1.61 ± 0.07 | 20.68 ± 2.60 | 174.41 ± 22.77 | 1931.95 ± 301.49 | 1.114 ± 0.020 | 20.68 ± 2.85 | 173.05 ± 33.01 | 1910.22 ± 272.73 | 577.37 ± 129.67 | 202.67 ± 28.71 |
| Cadical* | 0.90 ± 0.02 | 1.53 ± 0.08 | 22.76 ± 3.00 | 181.00 ± 25.32 | 1976.53 ± 293.76 | 0.864 ± 0.015 | 16.07 ± 2.16 | 173.71 ± 25.21 | 1874.66 ± 320.17 | 464.05 ± 74.14 | 271.77 ± 44.15 |
| Minisat (Graph-Q-Sat) | 33.96 ± 2.38 | 36.78 ± 2.11 | 80.53 ± 5.77 | 430.15 ± 43.52 | 2437.76 ± 286.59 | 0.083 ± 0.001 | 70.77 ± 5.25 | 183.71 ± 19.39 | 1737.40 ± 195.71 | 2992.97 ± 307.96 | 2971.47 ± 241.51 |
| Minisat* | 0.28 ± 0.05 | 1.31 ± 0.22 | 17.10 ± 2.02 | 63.96 ± 8.94 | 570.46 ± 87.94 | 0.259 ± 0.012 | 15.62 ± 2.21 | 60.73 ± 6.05 | 868.71 ± 81.62 | 43.69 ± 3.586 | 143.32 ± 22.89 |

BMM, MapleCOMSPS-BMM, MapleSat-BMM, and MapleLCMDist-BMM). The second category includes solvers with external learnable modules, such as NeuroBack [38] (based on kissat), Neuro-Cadical [15] (based on Cadical), and Graph-$Q$-Sat [10] (based on Minisat). For both categories, we used the publicly available official code and tested their average solving time per sample on the dataset. For solvers in the second category, we used their pre-trained models for testing.

We evaluated these ML-based heuristic enhancement methods across datasets derived from five different cryptographic algorithms: the block cipher Cipher mentioned in Section 5.1 with $k$ rounds and 32-bit key length, referred to as Cipher-k; the Simon block cipher [39] with $k$ rounds, n-bit key length and 2n-bit block size, denoted as Simon-n-2n; the Message Digest algorithm MD4 [40], the Secure Hash Algorithm SHA-256 [41], and the Secure Hash Algorithm SHA-1 [42].

The experimental results are shown in Table 2. Notably, the effectiveness of internal learnable modules varies with instance complexity and algorithm type. For simpler instances, traditional solvers without ML-based heuristic enhancements outperform their enhanced counterparts. As instance complexity increases, particularly in block cipher instances like Cipher-12 and Simon-12-32-64, solvers with ML-based enhancements show substantial advantages. For example, MapleLCMDist-BMM solves Cipher-12 in 368.26s, slightly outperforming its base version, MapleLCMDist, and significantly outperforming other traditional solvers. Besides, BMM-enhanced variants, such as MapleSat-BMM and MapleLCMDist-BMM, exhibit robust performance across different cryptographic families. The performance advantage of BMM-enhanced solvers suggests that learned branching heuristics can effectively capture and exploit structural patterns in cryptographic SAT instances.

In contrast, external learnable modules show mixed results. Neuro-Cadical performs similarly to its base solver, Cadical, in simpler instances, but the overhead from external learning increases in more complex cases. For example, Cadical outperforms Neuro-Cadical on Simon-12-32-64. NeuroBack and Graph-$Q$-Sat show even greater overhead, especially on SHA-1 and SHA-256. A likely reason is that their publicly available pre-trained models, such as NeuroBack, **fail to generalize well from their training sets to cryptographic problems, due to the latter's structures and larger scale.**

Table 3: Performance comparison of SAT solvers with different hyperparameter optimization methods on cryptographic instances. The results show average solving time in seconds, and we show the 95% confidence interval. For Kissat, the optimized parameters are (restartint,reduceint,decay). For Cryptominisat, the optimized parameters are (gluehist,rstfirst,confbtwsimp). For MapleSAT, the optimized parameters are (rnic,phase-saving,rnd-freq). The suffixes -H and -E denote HEBO and EasyNAS optimization methods, respectively.

| Solver | Simon-12-32-64 | Cipher-12 | MD4 | SHA-256 | SHA-1 |
|---|---|---|---|---|---|
| **Kissat** | 818.92 ± 55.93 | 974.35 ± 66.02 | 0.078 ± 0.001 | 293.78 ± 15.21 | 263.89 ± 14.56 |
| **Kissat-H** | 381.46 ± 20.10 | 521.42 ± 25.25 | 0.059 ± 0.001 | 276.55 ± 14.79 | 240.33 ± 14.70 |
| **Kissat-E** | 595.47 ± 30.67 | 583.55 ± 27.69 | 0.062 ± 0.001 | 278.85 ± 14.92 | 243.86 ± 13.05 |
| **Cryptominisat** | 3349.52 ± 115.28 | 3212.63 ± 114.04 | 0.076 ± 0.001 | 745.24 ± 54.18 | 227.27 ± 9.59 |
| **Cryptominisat-H** | 3027.48 ± 88.41 | 2894.72 ± 116.02 | 0.026 ± 0.001 | 615.73 ± 50.08 | 187.46 ± 10.21 |
| **Cryptominisat-E** | 3003.01 ± 88.62 | 2845.21 ± 115.53 | 0.024 ± 0.001 | 553.11 ± 40.25 | 185.29 ± 9.83 |
| **MapleSAT** | 440.02 ± 18.77 | 381.22 ± 15.94 | 1.435 ± 0.013 | 111.52 ± 14.41 | 75.92 ± 13.07 |
| **MapleSAT-H** | 378.14 ± 15.73 | 304.67 ± 12.97 | 0.425 ± 0.007 | 87.47 ± 6.91 | 60.42 ± 5.42 |
| **MapleSAT-E** | 371.78 ± 14.48 | 338.17 ± 14.89 | 0.276 ± 0.005 | 94.72 ± 8.31 | 69.87 ± 6.86 |

Overall, our results suggest that **internal ML-based heuristic modules are the more promising approach for improving solvers' performance on complex cryptographic instances. The small performance gap between ML-enhanced and traditional solvers on simpler instances underscores the need to balance ML benefits with its computational overhead.** Adaptive methods that activate ML components based on instance features may further enhance efficiency.

## 5.4 Evaluation of Hyperparameter Optimization

We conducted experiments to evaluate hyperparameter optimization strategies on cryptographic SAT instances, focusing on three widely used SAT solvers and two advanced optimization methods across diverse cryptographic algorithms.

**SAT Solvers and Parameters.** We employ three representative SAT solvers: Kissat [43], a state-of-the-art CDCL solver; CryptoMiniSat [44], specifically designed for cryptographic problems; and MapleSAT [36], known for its efficient branching heuristics. For each solver, we identified and optimized the most time-critical hyperparameters that directly influence solving efficiency.

**Optimization Methods** We implement two distinct approaches to optimize the hyperparameters: HEBO [45], a Heteroscedastic Evolutionary Bayesian Optimization method for black-box optimization, and EasyNAS [46], an evolutionary algorithm-based optimization approach. These methods were selected for their proven effectiveness in complex parameter spaces.

**Cryptographic Datasets** The evaluation framework includes five distinct cryptographic problems: the Simon block cipher with 12 rounds and 64-bit block size Simon-12-32-64 [39], Cipher-12, the Message Digest algorithm MD4 [40], the Secure Hash Algorithm SHA-256 [41], and the Secure Hash Algorithm SHA-1 [42]. Each cryptographic algorithm dataset comprises 210 instances, with 10 instances allocated for the training set and 200 instances for the testing set. The hyperparameter optimization was conducted using the training set, while performance evaluation was performed on the separate test set to ensure unbiased assessment.

Table 3 presents solver performance across cryptographic datasets under different hyperparameter optimization strategies. The results show that **ML-based hyperparameter optimization, particularly HEBO, can significantly improve solver efficiency**. For instance, Kissat achieved up to 62.5% reduction in solving time on the Cipher-12 dataset, and MapleSAT showed consistent gains

Table 4: Performance comparison of the best methods under different input representations on Simon cryptographic SAT instances. HO means the hyperparameter optimization. MVC is an external ML method in WDSat that guides variable enumeration by solving a Minimal Vertex Cover problem.

| Input Format | Simon-10-32-64 | | Simon-11-32-64 | | Simon-12-32-64 | |
|---|---|---|---|---|---|---|
| | Time (s) | Method | Time (s) | Method | Time (s) | Method |
| CNF Representation | 11.28 | Kissat | 41.57 | MapleSat+BMM | 371.78 | MapleSat+HO-EasyNAS |
| ANF Representation | **0.08** | WDSat | **23.60** | WDSat+MVC | 364.96 | WDSat+MVC |
| Plaintext-Ciphertext bitstring | 19.94 | BASIN | 37.30 | BASIN | **29.67** | BASIN(ours) |

across all datasets, with an 88.0% reduction on MD4. Cryptominisat also benefited substantially, with HEBO and EasyNAS each outperforming the other on specific datasets. While HEBO generally yielded better results, EasyNAS proved more effective in some block cipher settings, suggesting that the effectiveness of optimization strategies can be solver- and problem-specific. We also observed that block ciphers (e.g., Simon, Cipher-12) and hash functions (e.g., SHA-1, SHA-256) exhibited distinct optimization patterns, with MD4 showing unique behavior due to its structural differences.

## 5.5 Evaluation of Input Representations

To investigate the impact of input representations on solver performance, we conducted comparative experiments on Simon block cipher datasets of varying scales, using CNF, ANF, and plaintext-ciphertext bitstrings as input formats. For the CNF representation, we evaluated all aforementioned CNF-based solvers. For the ANF representation, we assessed WDSat [47], an ANF-based solver designed for algebraic normal form instances. For the plaintext-ciphertext bitstring representation, we evaluated our bitwise solver, BASIN, which performs key recovery through guided bitwise search. Table 4 summarizes the comparison of the best-performing methods under each representation. More comprehensive results for all ML-based solvers are provided in Appendix A.6.

As shown in Table 4, the bitwise solver **BASIN** achieves the best performance on higher-round instances, significantly outperforming CNF-based and ANF-based methods. CNF solvers like Kissat perform well on small instances but scale poorly. ANF better preserves algebraic structure, allowing WDSat to outperform CNF methods on early rounds, though it struggles with deeper ones. In contrast, the plaintext-ciphertext format enables BASIN to maintain efficiency on complex instances through bit-level reasoning and neural-guided initialization. These findings highlight the strong influence of input representation on solver performance, suggesting that **direct input representations, such as plaintext-ciphertext bitstrings, can more effectively capture the underlying cryptographic structures in complex cases.**

## 6 Conclusion

In this paper, we introduced SATCryptoBench, a benchmark for evaluating ML-assisted SAT solvers on cryptographic problems. Our analysis shows that input representation significantly impacts solver performance. ANF formats allow neural networks to better capture structural properties compared to CNF. Solvers leveraging domain-specific inputs, such as bit-level encodings from plaintext-ciphertext pairs, achieve superior efficiency on deep-round encryption tasks, highlighting the benefits of customized input formats. Besides, internal ML-based heuristic modules also outperform external guidance approaches in handling complex instances. Furthermore, hyperparameter tuning enhances traditional CDCL solvers by better aligning them with cryptographic instance characteristics.

Nonetheless, challenges remain. ML models often struggle to generalize across algorithms and instance scales, reflecting limitations in modeling cryptographic complexity. Besides, ML integration incurs overhead that may outweigh its benefits on simpler tasks. These findings underscore the need for specialized models for cryptographic structures. We hope SATCryptoBench will serve as a standardized resource to advance research in ML-based SAT solving and cryptanalysis.

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

# A  Experiment Setting

In this section, we provide a detailed description of our experimental methodology and settings.

## A.1  Computational Resources

All experiments were conducted on a high-performance computing server equipped with **Ascend 910B** AI processors. Training and inference tasks for neural network components were executed entirely on Ascend 910B GPUs using the PyTorch backend compiled for the Ascend platform. SAT solving components ran on CPU with 192 physical cores in 64-bit `aarch64` mode and NUMA-aware scheduling. Each task was allocated a dedicated Ascend device and CPU NUMA node to avoid inter-device contention.

## A.2  Datasets

For all cryptographic algorithms, we generated 1000 pairs for the training set, 200 pairs for the validation set, and 200 pairs for the test set. For Standalone Distinguisher evaluation, we generated datasets containing both ANF and CNF representations of the Simon family with 3 and 6 rounds, using block sizes of 8, 16, and 32 bits. For Heuristic Enhancement methods, we generated datasets for the Simon family with 10-12 rounds using a block size of 32 bits. Additionally, we generated datasets for Cipher-8 through Cipher-12, as well as MD4, SHA1, and SHA256. For Hyperparameter Optimization experiments, we conducted tests on Simon-12-32-64, Cipher-12, MD4, SHA1, and SHA256 datasets. To quantify the complexity of different cryptographic instances, we report the average number of variables and clauses for each dataset in Table 5.

Table 5: Dataset size statistics used in our experiments. We report the average number of variables (i.e., literals) and clauses.

| Dataset | Average #Variables | Average #Clauses |
|---|---|---|
| Simon-3-8-16-CNF | 120 | 392 |
| Simon-3-8-16-ANF | 16 | 24 |
| Simon-3-16-32-CNF | 240 | 784 |
| Simon-3-16-32-ANF | 32 | 48 |
| Simon-3-32-64-CNF | 360 | 1176 |
| Simon-3-32-64-ANF | 48 | 72 |
| Simon-6-8-16-CNF | 216 | 752 |
| Simon-6-8-16-ANF | 40 | 48 |
| Simon-6-16-32-CNF | 432 | 1504 |
| Simon-6-16-32-ANF | 80 | 96 |
| Simon-6-32-64-CNF | 648 | 2256 |
| Simon-6-32-64-ANF | 120 | 144 |
| Cipher-8-CNF | 608 | 4992 |
| Cipher-9-CNF | 672 | 5600 |
| Cipher-10-CNF | 736 | 6208 |
| Cipher-11-CNF | 800 | 6816 |
| Cipher-12-CNF | 864 | 7424 |
| MD4-CNF | 3504 | 37512 |
| SHA-1-CNF | 3968 | 119208 |
| SHA-256-CNF | 9070 | 151508 |
| Simon-10-32-64-CNF | 1376 | 4928 |
| Simon-11-32-64-CNF | 1504 | 5408 |
| Simon-12-32-64-CNF | 1632 | 5888 |
| Simon-10-32-64-ANF | 288 | 320 |
| Simon-11-32-64-ANF | 320 | 352 |
| Simon-12-32-64-ANF | 352 | 384 |

## A.3  Standalone Distinguisher

For all neural networks, we conducted training on the training set and evaluated the prediction accuracy on the test set. To ensure robust evaluation, we maintained strict separation between training and test sets, with no overlapping instances.

### A.4 Heuristic Enhancement

We categorized Heuristic Enhancement methods into two classes based on their learning module integration approach. The first class comprises solvers with internal learnable modules that dynamically adjust solving strategies during execution. For these methods, we conducted evaluations directly on the test set (using only SAT instances) and measured the average solving time. We set a timeout threshold of 5000 seconds, with instances exceeding this limit recorded as 5000 seconds in our statistics. The second class consists of solvers requiring pre-trained external neural network modules. For these methods, we first trained the neural networks on the training set and then evaluated their performance on the test set (using only SAT instances), measuring the average solving time with the same 5000-second timeout threshold.

### A.5 Hyperparameter Optimization

In our hyperparameter optimization experiments, we employed optimization methods to search for optimal parameters using the training set and evaluated the performance on the test set. For the Kissat solver, we optimized three hyperparameters: restartint, reduceint, and decay. For Cryptominisat, we focused on optimizing gluehist, rstfirst, and confbtwsimp. For MapleSat, we optimized rnic, phase-saving, and rnd-freq. Table 6 provides the specific parameter settings corresponding to the optimal combinations summarized in Table 3.

### A.6 Input Representations

To assess the impact of different input representations on solver performance, we generated datasets for CNF, ANF, and plaintext-ciphertext formats, and conducted comprehensive comparative experiments, as detailed in Table 7. The results show significant differences in solver performance depending on the input format. For example, plaintext-ciphertext input for our proposed BASIN solver outperforms both CNF- and ANF-based solvers on high-round instances. These findings underscore the importance of input representation in cryptographic SAT problem-solving.

Table 6: The detailed hyperparameter settings in the experiments of the Hyperparameter Optimization. For kissat, the optimized parameters are (restartint,reduceint,decay). For cryptominisat, the optimized parameters are (gluehist,rstfirst,confbtwsimp). For maplesat, the optimized parameters are (rnic,phase-saving,rnd-freq). -H and -E denote HEBO and EasyNAS optimization methods respectively.

| Solver | Simon-12-32-64 | Cipher-12 | MD4 | SHA-256 | SHA-1 |
|---|---|---|---|---|---|
| **Kissat** | (1,1000,50) | (1,1000,50) | (1,1000,50) | (1,1000,50) | (1,1000,50) |
| **Kissat-H** | (36,3258,1) | (42,3259,1) | (5767,98402,171) | (2,717,43) | (1,1099,1) |
| **Kissat-E** | (34,4258,2) | (73,5693,1) | (1790,20390,83) | (1,627,36) | (39,1944,6) |
| **Cryptominisat** | (50,100,40000) | (50,100,40000) | (50,100,40000) | (50,100,40000) | (50,100,40000) |
| **Cryptominisat-H** | (216,1633,99991) | (60,188,39988) | (232,1199,30976) | (437,374,48867) | (291,615,47226) |
| **Cryptominisat-E** | (216,1624,98636) | (73,288,39990) | (390,1920,21406) | (7,610,13156) | (293,639,47268) |
| **MapleSAT** | (1.5,0,0.02) | (1.5,0,0.02) | (1.5,0,0.02) | (1.5,0,0.02) | (1.5,0,0.02) |
| **MapleSAT-H** | (1359.74,1,0.001) | (2120.49,2,0) | (1917.07,0,0.001) | (410.39,0,0.009) | (2640.45,0,0.003) |
| **MapleSAT-E** | (1346.0,1,0.002) | (1963.8,0,0.001) | (2660.1,2,0.027) | (459.3,0,0.010) | (2558.2,0,0.005) |

Table 7: Detailed performance comparison of the best methods under different input representations on Simon cryptographic SAT instances. HO means the hyperparameter optimization. the default baseline solvers using their publicly available official implementations. MVC is an external ML method in WDSat that guides variable enumeration by solving a Minimal Vertex Cover problem.

| Input Format | Solver | Method | Dataset | Solving Time(s) |
|---|---|---|---|---|
| **CNF Representation** | Kissat | base | Simon-10-32-64 | 11.28 |
| | | base | Simon-11-32-64 | 73.24 |
| | | base | Simon-12-32-64 | 818.92 |
| | | NeuroBack | Simon-10-32-64 | 433.77 |
| | | NeuroBack | Simon-11-32-64 | 1728.98 |
| | | NeuroBack | Simon-12-32-64 | 3783.78 |
| | | HO-HEBO | Simon-12-32-64 | 381.46 |
| | | HO-EasyNAS | Simon-12-32-64 | 595.47 |
| | MapleSat | base | Simon-10-32-64 | 26.46 |
| | | base | Simon-11-32-64 | 73.85 |
| | | base | Simon-12-32-64 | 440.02 |
| | | BMM | Simon-10-32-64 | 14.83 |
| | | BMM | Simon-11-32-64 | 41.57 |
| | | BMM | Simon-12-32-64 | 460.57 |
| | | Crypto | Simon-10-32-64 | 15.98 |
| | | Crypto | Simon-11-32-64 | 58.34 |
| | | Crypto | Simon-12-32-64 | 717.31 |
| | | HO-HEBO | Simon-12-32-64 | 378.14 |
| | | HO-EasyNAS | Simon-12-32-64 | 371.78 |
| | MaplePainless | base | Simon-10-32-64 | 22.00 |
| | | base | Simon-11-32-64 | 77.10 |
| | | base | Simon-12-32-64 | 445.31 |
| | Glucose | base | Simon-10-32-64 | 14.52 |
| | | base | Simon-11-32-64 | 79.15 |
| | | base | Simon-12-32-64 | 1434.34 |
| | | BMM | Simon-10-32-64 | 27.33 |
| | | BMM | Simon-11-32-64 | 90.46 |
| | | BMM | Simon-12-32-64 | 1388.51 |

| | | | |
|---|---|---|---|
| | MapleCOMSPS | base | Simon-10-32-64 | 28.77 |
| | | base | Simon-11-32-64 | 125.78 |
| | | base | Simon-12-32-64 | 1467.97 |
| | | BMM | Simon-10-32-64 | 32.33 |
| | | BMM | Simon-11-32-64 | 110.60 |
| | | BMM | Simon-12-32-64 | 1248.37 |
| | MapleLCMDist | base | Simon-10-32-64 | 24.41 |
| | | base | Simon-11-32-64 | 69.08 |
| | | base | Simon-12-32-64 | 389.66 |
| | | BMM | Simon-10-32-64 | 24.10 |
| | | BMM | Simon-11-32-64 | 63.65 |
| | | BMM | Simon-12-32-64 | 382.07 |
| **CNF Representation** | Cadical | base | Simon-10-32-64 | 16.07 |
| | | base | Simon-11-32-64 | 173.71 |
| | | base | Simon-12-32-64 | 1874.66 |
| | | Neuro Cadical | Simon-10-32-64 | 20.68 |
| | | Neuro Cadical | Simon-11-32-64 | 173.05 |
| | | Neuro Cadical | Simon-12-32-64 | 1910.22 |
| | Minisat | base | Simon-10-32-64 | 15.62 |
| | | base | Simon-11-32-64 | 60.73 |
| | | base | Simon-12-32-64 | 868.71 |
| | | Graph $\mathcal{Q}$ Sat | Simon-10-32-64 | 70.77 |
| | | Graph $\mathcal{Q}$ Sat | Simon-11-32-64 | 183.71 |
| | | Graph $\mathcal{Q}$ Sat | Simon-12-32-64 | 1737.40 |
| | Cryptominisat | base | Simon-12-32-64 | 3349.52 |
| | | HO-HEBO | Simon-12-32-64 | 3027.48 |
| | | HO-EasyNAS | Simon-12-32-64 | 3003.01 |
| **ANF Representation** | WDSat | base | Simon-10-32-64 | 0.08 |
| | | base | Simon-11-32-64 | 23.70 |
| | | base | Simon-12-32-64 | 481.54 |
| | | MVC | Simon-10-32-64 | 0.08 |
| | | MVC | Simon-11-32-64 | 23.6 |
| | | MVC | Simon-12-32-64 | 364.96 |
| **Plaintext-Ciphertext bitstring** | BASIN(ours) | base | Simon-10-32-64 | 19.94 |
| | | base | Simon-11-32-64 | 37.30 |
| | | base | Simon-12-32-64 | 29.67 |

# B    Representation of Cryptographic SAT Instances

In this section, we present how cryptographic problems are modeled into datasets using CNF and ANF representations. Appendix B.1 describes how common logical operations in cryptographic algorithms are encoded as CNF and ANF formulas. Appendix B.2 further illustrates, taking the Simon cipher as an example, how to construct SAT datasets in both CNF and ANF formats for cryptographic problems.

## B.1    Representation of Common Cryptographic Operations

**ANF Representation of Common Cryptographic Operations**. The following demonstrates the conversion of five fundamental cryptographic operations into ANF:

- Circular Left Shift ($\lll$): A circular left shift is represented by the equation: $Y = X \lll b$, where the result of shifting $X = \overline{x_{k-1}x_{k-2}...x_0}$ $b$ positions to the left is: $Y = \overline{y_{k-1}y_{k-2}...y_0}$ with the following relations:

$$y_i + x_{(i+b) \mod k} = 0, \text{ for } i = 0, 1, ..., k-1 \tag{6}$$

- Circular Right Shift ($\ggg$): A circular right shift is represented by: $Y = X \ggg b$, where the result of shifting $X = \overline{x_{k-1}x_{k-2}...x_0}$ $b$ to the right is: $Y = \overline{y_{k-1}y_{k-2}...y_0}$ with the following relations:

$$y_i + x_{(i-b+k) \mod k} = 0, \text{ for } i = 0, 1, ..., k-1 \tag{7}$$

- Modular Addition with modulo $2^k$ ($\boxplus$): The Modular addition operation between $X = \overline{x_{k-1}x_{k-2}...x_0}$ and $Y = \overline{y_{k-1}y_{k-2}...y_0}$ is represented by: $Z = X \boxplus Y$. The addition is performed modulo 2 and the relations are:

$$z_0 + x_0 + y_0 = 0, \ c_0 + x_0 y_0 = 0$$
$$z_i + x_i + y_i + c_{i-1} = 0, \text{ for } i = 0, 1, ..., k-1 \tag{8}$$
$$c_i + x_i c_{i-1} + y_i c_{i-1} + i = 0, \text{ for } i = 1, ..., k-2$$

- Bitwise XOR ($\oplus$): The bitwise XOR operation between $X = \overline{x_{k-1}x_{k-2}...x_0}$ and $Y = \overline{y_{k-1}y_{k-2}...y_0}$ is represented by: $Z = X \oplus Y$ where the result $Z = \overline{z_{k-1}z_{k-2}...z_0}$ with the relations:

$$z_i + x_i + y_i = 0, \text{ for } i = 0, 1, ..., k-1 \tag{9}$$

- Bitwise AND ($\cdot$): The bitwise AND operation between $X = \overline{x_{k-1}x_{k-2}...x_0}$ and $Y = \overline{y_{k-1}y_{k-2}...y_0}$ is represented by: $Z = X \cdot Y$ where the result $Z = \overline{z_{k-1}z_{k-2}...z_0}$ with the relations:

$$z_i + x_i y_i = 0, \text{ for } i = 0, 1, ..., k-1 \tag{10}$$

**CNF Representation of Common Cryptographic Operations**. The derivation of CNF expressions for the fundamental cryptographic operations above follows, maintaining consistency with the previously defined notation:

- Circular Left Shift ($\lll$): The CNF representation of the circular left shift operation is:

$$\left(y_i \vee \neg x_{(i+b) \mod k}\right) \wedge \left(\neg y_i \vee x_{(i+b) \mod k}\right)$$
$$, \text{ for } i = 0, 1, ..., k-1 \tag{11}$$

- Circular Right Shift ($\ggg$): The CNF representation of the circular right shift operation is:

$$\left(y_i \vee \neg x_{(i-b+k) \mod k}\right) \wedge \left(\neg y_i \vee x_{(i-b+k) \mod k}\right)$$
$$, \text{ for } i = 0, 1, ..., k-1 \tag{12}$$

- Modular Addition with modulo $2^k$ ($\boxplus$):
- Bitwise XOR ($\oplus$): The CNF representation of the bitwise XOR operation is:

$$(\neg z_i \vee y_i \vee x_i) \wedge (z_i \vee \neg y_i \vee x_i) \wedge (z_i \vee y_i \vee \neg x_i) \wedge$$
$$(\neg z_i \vee \neg y_i \vee \neg x_i), \text{ for } i = 0, 1, ..., k-1 \tag{13}$$

- Bitwise AND ($\cdot$): The CNF representation of the bitwise AND operation is:

$$(z_i \vee \neg y_i \vee \neg x_i) \wedge (\neg z_i \vee y_i) \wedge (\neg z_i \vee x_i)$$
$$, \text{ for } i = 0, 1, ..., k-1 \tag{14}$$

## B.2 ANF and CNF Formula of Simon

In this section, we showcase the data generation process of both CNF and ANF formula using the Simon cipher as a concrete example. We begin with a concise description of the Simon encryption algorithm, followed by a detailed exploration of its ANF & CNF generation logic. Then, we demonstrate the conversion of the Simon encryption algorithm to SAT representations in both ANF and CNF forms.

**Simon Cipher Overview**: Simon [39] is a lightweight block cipher family utilizing a Feistel structure with variable block sizes, key sizes, and round numbers. The encryption process operates on two state halves $(L_i, R_i)$, following these round transformations:

$$L_{i+1} = ((L_i \lll 1) \cdot (L_i \lll 8)) \oplus (L_i \lll 2) \oplus R_i \oplus K_r$$
$$R_{i+1} = L_i \tag{15}$$

where $K_r$ represents the round key, $\lll$ denotes a left circular shift, $\cdot$ represents the bitwise AND operation, and $\oplus$ denotes the bitwise XOR operation.

**Notation**: In the following explorations on ANF and CNF representations, we consider Simon configurations with a $2n$–bit block size implementing $r$ encryption rounds and using an $n$-bit seed key. Our analysis constructs symbolic equations from known plaintext-ciphertext pairs, where the encryption key remains the unknown variable to be determined.

The core of Simon's encryption algorithm lies in representing its round structure in ANF and CNF forms. As shown in Equation 15, Simon's round function consists of only three basic operations: circular left shift, bitwise XOR, and bitwise AND. Let $L_i = \overline{x_0, x_1, \ldots, x_{n-1}}$ and $R_i = \overline{y_0, y_1, \ldots, y_{n-1}}$ denote the two halves of the round function's input, where $x_i, y_i \in \{0, 1\}$. The round function produces output $(L_{i+1}, R_{i+1})$ with $L_{i+1} = \overline{u_0, u_1, \ldots, u_{n-1}}$ and $R_{i+1} = \overline{v_0, v_1, \ldots, v_{n-1}}$, where $u_i, v_i \in \{0, 1\}$.

**ANF formula for Simon's round function**: We can directly transform the Simon round function into an ANF-based SAT instance following the conversion method outlined in Appendix B.1:

$$u_i + x_{i+1 \mod n} \cdot x_{i+8 \mod n} + x_{i+2 \mod n} + y_i + k_i = 0$$
$$v_i + x_i = 0 \tag{16}$$

**CNF formula for Simon's round function**: Unlike the ANF representation, the CNF representation of Simon requires introducing intermediate variables for simplification. Let $A_1^i = (L_i \lll 1) \cdot (L_i \lll 8)$, $A_2^i = A_1^i \oplus (L_i \lll 2)$, $A_3^i = A_2^i \oplus R_i$, where $A_j^i = \overline{a_0^j, a_1^j, \ldots, a_{n-1}^j}$. Then, the round function of the Simon encryption algorithm can be transformed into the following set of CNF clauses:

$$
\begin{aligned}
(\neg a_m^1 \vee x_{m+1 \mod n}) \wedge (\neg a_m^1 \vee x_{m+8 \mod n}) \wedge & \\
(a_m^1 \vee \neg x_{m+8 \mod n} \vee \neg x_{m+1 \mod n}) & \\
, \text{for } m = 0, 1, ..., n-1 &
\end{aligned}
\tag{17}
$$

$$
\begin{aligned}
(\neg a_m^2 \vee a_m^1 \vee x_{m+2 \mod n}) \wedge (a_m^2 \vee \neg a_m^1 \vee x_{m+2 \mod n}) \wedge & \\
(a_m^2 \vee a_m^1 \vee \neg x_{m+2 \mod n}) \wedge (\neg a_m^2 \vee \neg a_m^1 \vee \neg x_{m+2 \mod n}) & \\
, \text{for } m = 0, 1, ..., n-1 &
\end{aligned}
\tag{18}
$$

$$
\begin{aligned}
(\neg a_m^3 \vee a_m^2 \vee y_m) \wedge (a_m^3 \vee \neg a_m^2 \vee y_m) \wedge & \\
(a_m^3 \vee a_m^2 \vee \neg y_m) \wedge (\neg a_m^3 \vee \neg a_m^2 \vee \neg y_m) & \\
, \text{for } m = 0, 1, ..., n-1 &
\end{aligned}
\tag{19}
$$

$$
\begin{aligned}
(\neg u_m \vee a_m^3 \vee k_m) \wedge (u_m \vee \neg a_m^3 \vee k_m) \wedge & \\
(u_m \vee a_m^3 \vee \neg k_m) \wedge (\neg u_m \vee \neg a_m^3 \vee \neg k_m) & \\
, \text{for } m = 0, 1, ..., n-1 &
\end{aligned}
\tag{20}
$$

$$(v_m \vee \neg x_m) \wedge (\neg v_m \vee x_m), , \text{for } m = 0, 1, ..., n-1 \tag{21}$$

## C  XOR operation bottleneck in CNF

Cryptographic algorithms often involve operations like XOR and modular addition with modulo $2^8$, making XOR operations common in SAT-based cryptanalysis. However, due to the differences in logical properties between XOR and CNF, representing XOR operations in the Conjunctive Normal Form (CNF) presents a significant bottleneck and makes the conversion computationally expensive.

CNF consists of a conjunction (AND) of clauses, where each clause is a disjunction (OR) of positive and negative variables, called literals. The flexibility of the "AND" and "OR" constraints in CNF enables efficient SAT-solving applications. A direct representation of an XOR clause involving $k$ literals, such as $x_1 \oplus x_2 \oplus \cdots \oplus x_k$, typically requires a large number of OR clauses, as Fig 3(a) shows. Without introducing intermediate variables, this conversion results in an exponential blow-up in the number of required OR clauses, specifically on the order of $2^{k-1} - 1$ OR clauses for a clause involving $k$ literals. This rapid increase in complexity makes direct XOR-to-CNF transformation impractical for large $k$.

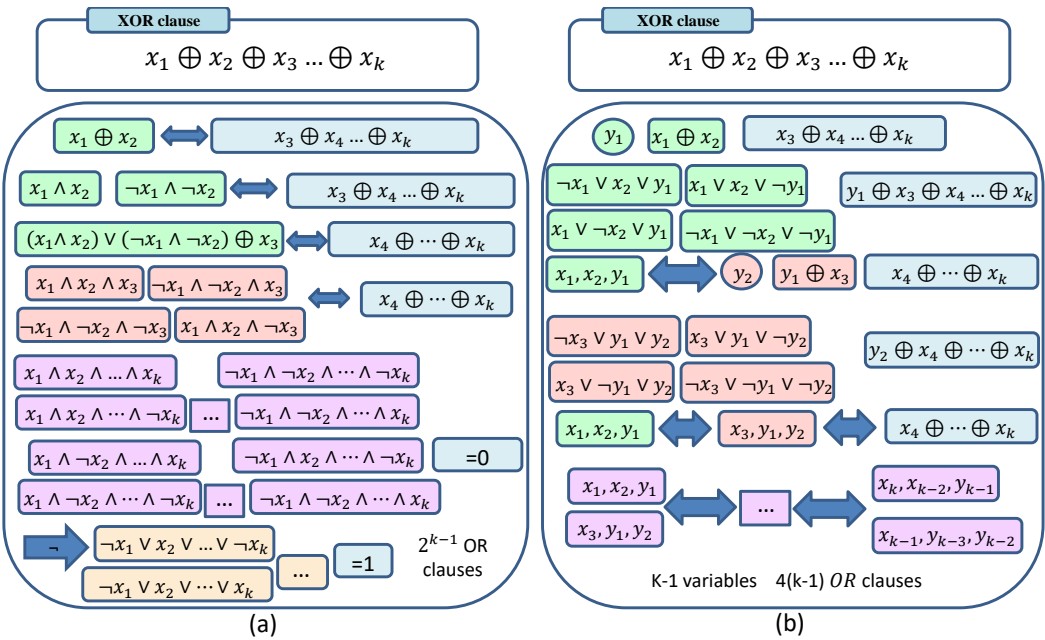

Figure 3: Two representations of an XOR clause involving $k$ literals. The circles represent intermediate variables, rectangles represent clauses. (a) Direct representation of the XOR clause, where different colors indicate that the clauses originate from different transformation steps. For an XOR clause involving $k$ literals, there exist a total of $2^k$ different values of $(x_1, \ldots, x_k)$. When considering the equation $x_1 \oplus x_2 \oplus \cdots \oplus x_k = 1$, there are $2^{k-1}$ possible solutions. By finding all solutions to the equation $x_1 \oplus x_2 \oplus \cdots \oplus x_k = 0$ and negating them, we obtain a CNF consisting of $2^{k-1}$ OR clauses. (b) Introduce intermediate variables, where variables and clauses of the same color come from the same XOR clause. The additional variables (denoted $y_1, y_2, \ldots, y_{k-1}$) are used to transform the XOR clause into $4(k-1)$ OR clauses, involving $k-1$ variables.

Another representation is to introduce intermediate variables that decompose the XOR operation into simpler ternary XOR clauses as Fig 3(b) shows. By defining new variables such as $y_1 = x_1 \oplus x_2$, $y_2 = y_1 \oplus x_3$, and so on, we reduce the XOR operation to a series of ternary XOR clauses. Each ternary XOR clause typically requires 4 OR clauses, resulting in a much smaller number of OR clauses—approximately $4(k-1)$ compared to the direct conversion. However, this transformation introduces numerous intermediate variables, still making the number of clauses and literals in SAT samples under the cryptographic problem in CNF difficult to solve.

# D  Limitations

The primary limitation of this paper is its focus on problem formulation and empirical analysis, without proposing concrete solutions. While the proposed benchmark and experiments reveal several important insights—such as the superiority of ANF representations and the sensitivity of solver performance to input formats—the study does not introduce or validate new solver architectures or optimization methods that can directly overcome the current limitations of ML-enhanced solvers, including poor generalization across algorithms and instance sizes, and the potential computational overhead outweighing benefits on simpler problems. As such, this work is intended to serve as a foundation to stimulate and guide future research, rather than providing complete solutions to the identified challenges. Addressing these challenges will require furture efforts to develop more adaptive, lightweight, and generalizable ML techniques, alongside deeper exploration of input representation learning.

# E  Impact Statements

The introduction of SAT4CryptoBench offers significant positive societal impacts by standardizing the evaluation of machine learning-based cryptanalysis tools, enhancing the development of robust

cryptographic methods that can proactively identify vulnerabilities. This contributes to stronger security protocols, safeguarding critical infrastructure and personal privacy. However, the research also poses risks, as advanced cryptanalytic techniques could be misused by malicious actors, undermining cryptographic protections. Additionally, the computational complexity of these methods may exacerbate disparities in cybersecurity capabilities, leaving less-resourced entities more vulnerable to sophisticated attacks. Thus, while promoting security advancements, the tool also requires cautious application to mitigate potential misuse.

