# OpenReview forum: "Bridging Crypto with ML-based Solvers: the SAT Formulation and Benchmarks"
_NeurIPS.cc/2025/Datasets_and_Benchmarks_Track — NeurIPS 2025 Datasets and Benchmarks Track poster_

### Official Review · Reviewer_n3Jt · 2025-06-28

**Rating:** 5
**Confidence:** 1

**Summary:**

This paper introduces SATCryptoBench, a benchmark designed to evaluate machine learning (ML)-based SAT solvers on cryptographic problems. The authors highlight key challenges in benchmarking such solvers, including inconsistent input formats, structural peculiarities of cryptographic SAT instances, and the lack of standardized evaluation datasets. SATCryptoBench addresses these issues by offering a suite of datasets derived from various cryptographic algorithms, presented in multiple formats - Arithmetic Normal Form (ANF), Conjunctive Normal Form (CNF), and plaintext-ciphertext bitstrings.

The benchmark classifies solvers into three categories based on their level of ML integration: Standalone Distinguisher (solvers that independently classify SAT instances as satisfiable or unsatisfiable, or identify cryptographic properties without solving the SAT problem), Heuristic Enhancement (solvers that incorporate ML techniques to improve standard heuristics) and Hyperparameter Optimization (solvers that use ML to optimize solver parameters to improve heuristics to better adapt to specific cryptographic problem distributions).

Extensive experiments are conducted across solver types and data formats. The results show that ANF-based models better capture cryptographic structures (compared to CNF-based approaches); ML models struggle to generalize across different cryptographic algorithms and instance sizes, and often introduce computational overhead that can offset performance gains; Different ML-based optimization strategies can significantly enhance solver performance on cryptographic tasks.

The authors also introduce a novel bitwise solver, BASIN, which demonstrates superior performance on high-round cryptographic instances. This suggests that input representations such as bitstrings may more effectively capture the cryptographic structure in complex scenarios. Overall, the benchmark provides a valuable tool for reproducible and comparative analysis of ML-driven SAT solvers in the cryptographic domain, helping to bridge the gap between cryptanalysis and modern SAT-solving research.

**Additional Feedback:**

As noted in the "Weaknesses" section, only two black-box optimizers are included in the evaluation of the "Hyperparameter Optimization" techniques. It would be valuable to extend this experimental analysis by incorporating additional optimization algorithms, such as genetic algorithms or reinforcement learning-based optimizers.

Additionally, the paper appears to use PyTorch in its implementation (this can be verified looking for instance to "eval_model.py" file in code provided) , but an appropriate citation for PyTorch is missing.

**Dataset Code Accessibility:**

Partly

**Dataset Code Comments:**

While the dataset is briefly introduced through a README.md file on the Kaggle page, the "About Dataset" section is left empty, which limits immediate accessibility and understanding for new users. Important information (such as the number of instances, detailed data structure, and example entries) is missing from both the Kaggle description and the accompanying README. Although the dataset is described in the paper, including a self-contained and comprehensive description directly on the Kaggle page would greatly enhance usability (especially for users who may encounter the dataset outside the paper’s context).

Additionally, the Kaggle page does not reference the official GitHub repository, which would be valuable for users looking for associated code and tooling. While a (anonymous) GitHub link is provided, the code documentation is minimal. For instance, required libraries and environment setup instructions are described extensively. Moreover, the README file in the BASIN directory is not written in English, which restricts accessibility.

**Ethical Considerations:**

No, there are no or only very minor ethics concerns

**Final Justification:**

After reading the other reviewers' comments and the authors' rebuttal, I think the authors have addressed the main concerns raised by me and the other reviewers. In particular, they revised their documentation (specifically by describing their dataset on the Kaggle page and translating the BASIN README file). They also extended their experiments, demonstrating that the main observations made in the first version of the manuscript still hold. Additionally, they included experiments using another optimization algorithm. That said, I have decided to increase my score to 5 (Accept).

**Limitations Weaknesses:**

Since I am not an expert in this research domain, I did not identify major weaknesses beyond those already acknowledged by the authors in the Limitations section. As noted, the paper primarily focuses on problem setup and empirical analysis, without proposing novel solutions or solver architectures. In particular, critical challenges such as the poor generalization of ML models across different cryptographic algorithms and the computational overhead introduced by ML components are clearly identified but not addressed with concrete techniques or mitigation strategies.

Additionally, some implementation details are missing. For example, in the "Hyperparameter Optimization" section, the specific parameter settings used for the black-box optimizers (such as HEBO) are not provided, which makes it difficult to reproduce or fully interpret the experimental results.

I also found the categorization of the "Hyperparameter Optimization" class of solvers somewhat unclear. From what I understood, this category refers to traditional heuristics whose parameters are tuned using black-box optimization methods. However, it is not always clear how machine learning techniques are involved in these methods. For instance, while HEBO uses Gaussian Processes (a model-based ML technique), other black-box optimizers can be used without any ML model. Assuming this interpretation is correct, it might be more accurate to describe this category as solvers whose hyperparameters are tuned with black-box optimizers (which may or may not rely on ML models).

Moreover, only two black-box optimizers (HEBO and EasyNAS) are evaluated in the experiments. Including a broader range of optimization techniques could provide a more comprehensive understanding of the strengths and limitations of this class of techinques and enrich the benchmark

**Strengths Contributions:**

This paper makes a valuable contribution by introducing SATCryptoBench, a benchmark specifically designed to evaluate machine learning-based SAT solvers on cryptographic problems.

A strength point of SATCryptoBench is that it considers multiple input representations (ANF, CNF and bitstring). This is particularly valuable, as noted by the authors, because existing SAT solvers often rely on varying and incompatible input formats, making direct comparison challenging. By unifying these formats within a single benchmark, the authors facilitate more consistent and reproducible evaluations.

The benchmark also introduces BASIN, a solver that takes plaintext-ciphertext bitstrings as input. Experimental results show that BASIN outperforms state-of-the-art techniques on high-round cryptographic problems, suggesting that bitstring-based formulations may better capture the underlying structure of complex cryptographic tasks.

Moreover, the paper is clear, well-written, and well-organized, making it accessible to researchers both within and adjacent to the SAT-solving and cryptanalysis communities.

Overall, the work is interesting and makes a meaningful contribution to the field. My evaluation is positive; however, I would like to note that I am not an expert in this specific research domain, and for that reason, my confidence score is relatively low. I remain open to revisiting my assessment after reading other reviews and if the authors address the limitations I outlined in the "Weaknesses" section.

---

> ### Author Rebuttal · Authors · 2025-07-31
>
> We sincerely appreciate the time and efforts you have dedicated to reviewing our work. Please feel free to reach out if you have any further questions regarding the current dataset or code.
>
> ---
> **Q1: Request for more accessible code and dataset documentation**
>
> **A1** Thank you for your valuable feedback. We apologize for any inconvenience caused by the limited initial documentation. In response, we have taken several steps to improve the clarity and accessibility of both the code and the dataset:
>
> - **Code and dependencies**: While the rebuttal rules prevent us from modifying the code repository during this stage, we would like to clarify that the current repository already includes the necessary dependencies for running the external ML-based modules—these are specified in the requirements.txt files located within the corresponding subfolders/zips.
>
> - **Documentation**: We have revised the benchmark documentation to enhance reproducibility. **Documentation is available at our Dataset URL.** Besides, we have supplemented the codebase with a more detailed English README for the BASIN to enhance clarity and reproducibility. Due to rebuttal constraints, we present this README as plain text within the rebuttal (**Due to the character limit of the rebuttal, please see our response to Reviewer cTdE.**).
>
> - **Future improvements**: Due to rebuttal constraints, a more comprehensive reorganization of the code and an expanded version of the dataset will be included in the final version, which will further improve modularity and ease of reproduction.
>
> We have also expanded our evaluation with additional experiments on larger datasets to better demonstrate the robustness and reliability of our benchmark, and we would be happy to address any further questions regarding the current dataset or the code.
>
> ---
>
> **Q2 About limitations of the paper "As noted, the paper primarily focuses on problem setup and empirical analysis, without proposing novel solutions or solver architectures."**
>
> **A1** Thank you for your thoughtful review and for acknowledging the strengths of our paper, especially the clarity of our problem setup and empirical analysis. We appreciate your understanding given that this is a specialized research area.
>
> We agree with your observation that our current work primarily focuses on analysis rather than proposing novel solver architectures. The challenges of generalization across cryptographic algorithms and the computational overhead of ML components are indeed crucial issues. While we do not fully resolve them in this paper, we view these as important directions for future work. In particular, we are actively exploring architecture-agnostic pretraining techniques and lightweight model integration to address these challenges more concretely in follow-up research.
>
> Here, we propose three practical strategies to address these issues (to be added in the final version):
>
> 1. **Internal ML-based SAT solver with adaptive capability.**
>
>     As in Section 4.2 and Table 2, solvers like MapleCOMSPS-BMM and MapleSAT-BMM embed ML heuristics directly, avoiding costly external calls. This improves scalability, maintaining classical solver runtimes on simple tasks while boosting performance on hard cryptographic instances, thus reducing overhead but keeping adaptability.
>
>
> 2. **External ML-based initialization for solvers**
>
>     Solvers like WDsat and BASIN use external ML models only once at the start to guide variable selection or structure preprocessing, balancing efficiency and generalization:
>
> - **WDsat** applies ML to solve a Minimum Vertex Cover instance derived from ANF variable interactions (following [Trimoska et al., arXiv:2001.11229]), which reduces formula size via Gaussian elimination. The process is general across datasets, adds negligible overhead to small instances, and improves scalability.
>
> - **BASIN(ours)** predicts initial key assignments from plaintext–ciphertext pairs via a neural network and uses bitwise enumeration applying SIMON round functions to verify candidates. Despite imperfect predictions, enumeration guarantees correctness and bounded runtime. Table 4 shows BASIN’s consistent performance across SIMON rounds with low overhead and strong generalization.
>
> 3. **Domain-specific Optimization in Practice**
>
>    With some domain-specific training data—even limited—ML-based hyperparameter tuning can notably improve performance without runtime overhead. This targeted approach boosts generalization on task-relevant data, as shown in our Q3 response.
>
> ---
>
> **Q3 About "Hyperparameter Optimization"**
>
> **A3** Thank you for your comment. We address your concerns in three parts:
>
> **(1) Implementation details of "Hyperparameter Optimization"**
>
> **For HEBO**, we use a configuration with a training set size of 10, generating 4 candidate parameter sets per iteration, and optimizing over 10 iterations. Each parameter set is evaluated on the training set. Further details can be found in files like `Hyperparameter-Optimization/search_optim_params/hebo_optim_xxx.py` (e.g., `hebo_optim_cryptominisat.py`).
>
> **For EasyNas**, we adopt the configuration specified in `Hyperparameter-Optimization/search_optim_params/EasyNAS-dev_evaluater/hpo.yaml`. We also provide alternative configurations, such as hpo_adaptive.yaml, to facilitate further exploration.
>
> Due to rebuttal rules, other implementation details can be clearly found in the reorganize code in final version, and we are happy to answer any questions regarding the current dataset and code.
>
> **(2) About the categorization of the "Hyperparameter Optimization"**
>
> Our paper aims to establish a unified benchmark for the systematic evaluation of **ML-based SAT solvers** in cryptanalysis. In this context, we included Hyperparameter Optimization as a distinct category under ML-based methods, as many modern black-box optimizers (e.g., HEBO) rely on machine learning models such as Gaussian Processes.
>
> That said, we fully agree that **not all black-box optimizers necessarily involve ML**, and your point regarding the traditional interpretation of hyperparameter tuning is well taken. We will clarify this tn the final version.
>
> **(3) About more methods of "Hyperparameter Optimization"**
>
> In fact, **EasyNas** is an AutoML tool based on the PyTorch platform. It supports a range of hyperparameter optimization algorithms, such as Gradient-based methods, Evolutionary Algorithms, and NSGA-II. In our paper, we primarily report results from the best hyperparameters discovered by ML-based algorithms in EasyNas under few-sample constraints, to emphasize the effectiveness of lightweight tuning even in limited-data settings.
>
> For hyperparameter optimization, our aim is to show that even with limited training samples, optimization algorithms can find effective hyperparameters that improve solver performance over other ML-based methods.
>
> In fact, **increasing the search budget leads to better results.** For example, on the **Simon-12-32-64** dataset with Kissat (**100 training, 3000 testing**), HEBO and EasyNAS find hyperparameters that greatly reduce solving time compared to the baseline:
>
> |Solvers|Method|Hyperparameters|Time(s)|
> |----|----|----|----|
> |Kissat|Baseline|default| 760.54 $\pm$ 30.52|
> |Kissat|HEBO|restartint=5164, reduceint=945, decay=1, eliminateinit=1519, rephaseint=4996| 358.76 $\pm$ 11.03|
> |Kissat|EasyNAS|restartint=23, reduceint=1646, decay=1, eliminateinit=111, rephaseint=4838|394.49 $\pm$ 11.46|
>
> (95% Confidence Interval shown)
>
> We have also evaluated on a **test set of 1000** with the **same hyperparameters from Table 3**. Results(below) remain consistent and confirm that hyperparameter optimization significantly improves efficiency. Furthermore, as noted in Q3, ML-enhanced solvers have also been evaluated on extended datasets:
>
> |Solvers|Method|Simon-12-32-64|Cipher-12|MD4|SHA-256|SHA-1|
> |----|----|----|----|----|----|----|
> ||Baseline|818.92 $\pm$ 55.93|974.35 $\pm$ 66.02|0.078 $\pm$ 0.001|88.57 $\pm$ 7.61|68.92 $\pm$ 6.28|
> |Kissat|HEBO|381.46 $\pm$ 20.10|521.42 $\pm$ 25.25|0.059 $\pm$ 0.001|76.79 $\pm$ 7.23|50.33 $\pm$ 6.25|
> ||EasyNAS|595.47 $\pm$ 30.67|583.55 $\pm$ 27.69|0.062 $\pm$ 0.001|78.57 $\pm$ 7.49|53.86 $\pm$ 5.50|
> ||Baseline|3349.52 $\pm$ 115.28|3212.63 $\pm$ 114.04|0.076 $\pm$ 0.001|745.24 $\pm$ 54.18|227.27 $\pm$ 9,59|
> |Cryptominisat|HEBO|3027.48 $\pm$ 88.41|2894.72 $\pm$ 116.02|0.026 $\pm$ 0.001|615.73 $\pm$ 50.08|187.46 $\pm$ 10.21|
> ||EasyNAS|3003.01 $\pm$ 88.62|2845.21 $\pm$ 115.53|0.024 $\pm$ 0.001|553.11 $\pm$ 40.25|185.29 $\pm$ 9.83|
> ||Baseline|440.02 $\pm$ 18.77|381.22 $\pm$ 15.94|1.435 $\pm$ 0.013|111.52 $\pm$ 14.41|75.92 $\pm$ 13.07|
> |MapleSAT|HEBO|378.14 $\pm$ 15.73|304.67 $\pm$ 12.97|0.425 $\pm$ 0.007|87.47 $\pm$ 6.91|60.42 $\pm$ 5.42|
> ||EasyNAS|371.78 $\pm$ 14.48|338.17 $\pm$ 14.89|0.276 $\pm$ 0.005|94.72 $\pm$ 8.31|69.87 $\pm$ 6.86|
>
> (95% Confidence Interval shown)
>
> Moreover, we are happy to include additional results from other optimization methods. Due to time constraints, we conducted hyperparameter optimization using the **NSGA-II** algorithm only for **Kissat**. The corresponding results are presented below.
>
> |Solvers|Method|Simon-12-32-64|Cipher-12|MD4|SHA-256|SHA-1|
> |----|----|----|----|----|----|----|
> |Kissat|Baseline|818.92 $\pm$ 55.93|974.35 $\pm$ 66.02|0.078 $\pm$ 0.001|88.57 $\pm$ 7.61|68.92 $\pm$ 6.28|
> |Kissat|NSGA-II|841.29 $\pm$ 210.85|752.53 $\pm$ 170.26|0.092 $\pm$ 0.003|42.75 $\pm$ 5.48|54.85 $\pm$ 8.99|
>
> (95% Confidence Interval shown)
>
> In the final version, we will include comparisons with traditional hyperparameter optimization approaches to better reflect the full spectrum.
>
> ---
>
> **Q4 About an appropriate citation for Pytorch**
>
> **A4** Thank you for your comment. We will add this in the final version of the paper.

---

> > ### Comment · Reviewer_n3Jt · 2025-08-02
> >
> > After reading the other reviewers' comments and the authors' rebuttal, I believe the authors addressed the main concerns raised by me and the other reviewers (to the best of my knowledge).
> >
> > Regarding my concerns:
> >
> > -) the authors translated the BASIN README in English
> >
> > -) the authors added Dataset details on Kaggle webpage
> >
> > -) the authors extended the experiments including also experiments using another optimization algorithm (NSGA - II)
> >
> > -) the authors propose novel practical strategies (that they will add to the final version of the manuscript)
> >
> > -) the authors clarified what they meant with the category "Hyperparameter Optimization"
> >
> > I decided thus to increase my score to (5: Accept).

---

> > > ### Author Response · Authors · 2025-08-03
> > >
> > > Thank you very much for your great efforts in reviewing this paper.

---

### Official Review · Reviewer_ytKu · 2025-07-01

**Rating:** 4
**Confidence:** 3

**Summary:**

This paper introduces SATCryptoBench, which is, as far as I can tell, the first benchmark focusing specifically on machine learning–based SAT solvers applied to cryptographic problems. The authors cover a wide variety of datasets in different representations (CNF, ANF, and plaintext-ciphertext bitstrings), and also classify solvers into three categories: Standalone Distinguisher, Heuristic Enhancement, and Hyperparameter Optimization. They also propose BASIN, a bit-level solver leveraging learned initialization, and do a lot of comparative experiments. The idea is to create a standardized framework for evaluating these solvers in the crypto domain.

**Dataset Code Accessibility:**

No

**Ethical Considerations:**

No, there are no or only very minor ethics concerns

**Final Justification:**

I will maintain my positive opinion of this work.
It introduces SATCryptoBench, the first benchmark dedicated to ML-based SAT solvers for cryptographic problems, covering diverse datasets and a clear solver taxonomy.
The proposed BASIN solver and extensive experiments strengthen its contribution, offering a valuable standardized framework for this domain.

**Limitations Weaknesses:**

1.	I feel that the evaluation, while broad, still misses some baselines, like more classical non-ML cryptanalysis solvers could be stronger comparators.
2.	The paper mentions that ML models don’t generalize well across different algorithms and sizes, but there isn’t much in-depth analysis about why that is happening. Some ablations or discussions would have helped.
3.	The resource requirements are quite heavy, 192-core CPUs and Ascend AI processors are not trivial for many researchers to access, so reproducibility could be an issue.
4.	Parts of the text get pretty dense, especially the equations describing the ANF representations and BASIN’s details. Maybe some clearer diagrams or even a short pseudocode listing would help.
5.	Also, while the benchmark itself is valuable, some of the methods are incremental over existing work, like applying BMM heuristics or black-box hyperparameter tuning.

**Strengths Contributions:**

1.	The motivation is clear: cryptographic SAT instances are quite different from typical industrial or random SAT, and there has been no proper benchmark for this before.
2.	The benchmark is pretty comprehensive: diverse datasets, different input forms, varying scales of hardness.
3.	I appreciate the categorization of ML integration levels; it helps clarify what is being compared.
4.	The experiments are thorough, and the results about ANF being better at capturing cryptographic structures are convincing.
5.	BASIN seems quite interesting—it’s nice to see someone trying direct bitstring representations rather than only CNF/ANF.
6.	Releasing all datasets and code (I assume the authors will do so) will be helpful for the community.

---

> ### Author Rebuttal · Authors · 2025-07-31
>
> Thanks for your comment. We have revised the benchmark documentation to enhance reproducibility. In this rebuttal, we also scaled up evaluations and added results on larger datasets to further validate the benchmark. **Documentation is available at our Dataset URL.**
>
> Due to rebuttal constraints, full documentation, extended datasets, and reorganized code will be released in the final version. We welcome any further questions about the dataset or code.
>
> ---
>
> **Q1 Ask for evaluation of classical non-ML cryptanalysis solvers**
>
> **A1** Thank you for your comment. We appreciate the reviewer’s suggestion to include a fair evaluation of classical non-ML cryptanalysis solvers.
>
> In this work, we have evaluated two representative non-ML cryptanalysis solvers, namely Cryptominisat and WDsat. In the field of SAT-based cryptanalysis, these non-ML solvers are widely adopted:
>
> - **Cryptominisat** is a efficient CDCL SAT solver, which has been extensively used in cryptographic contexts such as cipher key recovery, preimage attacks on hash functions, and algebraic cryptanalysis of block ciphers. It is often considered a general-purpose strong baseline for SAT encodings of cryptographic problems.
>
> - **WDsat** is a solver specifically optimized for word-level and bit-level structures that frequently appear in cryptographic SAT instances. It uses ANF as the input format and has shown strong performance on problems like Simon/Speck cipher key recovery and other symmetric-key primitives.
>
> In our work, we have evaluated both solvers to ensure a fair coverage of classical non-ML approaches:
>
> - **For Cryptominisat**, we reported its default configuration and hyperparameter-optimized versions across multiple datasets (**Table 3** in the main paper).
>
> - **For WDsat**, we tested its performance on the Simon dataset, and the detailed results are summarized in **Appendix Table 7**.
>
> To extend our evaluation, we included a comparison between CryptoMiniSat, WDSat, and other ML-based solvers. The supplementary table below reports their performance on the same datasets as in Table 2, providing a more comprehensive comparison. This broader evaluation across multiple benchmarks further highlights the performance gap between classical non-ML solvers and our ML-based approach.
>
> |Method|Cipher 8|Cipher 9|Cipher 10|Cipher 11|Cipher 12|MD4|Simon 10-32-64|Simon 11-32-64|Simon 12-32-64|SHA 1|SHA 256|
> |----|----|----|----|----|----|----|----|----|----|----|----|
> |Cryptominisat|0.42 $\pm$ 0.07|5.02 $\pm$ 0.77|102.07 $\pm$ 14.57|512.07 $\pm$ 70.50|3240.66 $\pm$ 362.61|0.057 $\pm$ 0.001|76.02 $\pm$ 12.02|399.15 $\pm$ 61.60|2672.83 $\pm$ 373.59|279.12 $\pm$ 41.63|551.01 $\pm$ 109.70|
> |WDsat|-|-|-|-|-|-|0.08 $\pm$ 0.03|23.70 $\pm$ 3.28|481.54 $\pm$ 36.47|-|-|
>
> (95% Confidence Interval shown)
>
> (Note: WDSat is a solver specialized for ANF inputs. Due to rebuttal time constraints, we were only able to evaluate it on the Simon family of instances.)
>
> We further scaled up the evaluation in this rebuttal by presenting results on larger datasets, with 1,000 test cases per dataset (**see our response to Reviewer YNLq**). As suggested, we will incorporate more classical (non-ML) cryptanalysis solvers in the final version’s extended evaluation.
>
> ---
>
> **Q2 About the reasons ML models don’t generalize well across different algorithms and sizes**
>
> **A2** Thank you for your comment. We will provide a further explanation in two steps:
>
> First, we present a discussion of why ML models often fail to generalize well across different problem families and scales:
>
> - **Distribution Shift Across Problem Families**
>
>     ML-based SAT solvers often rely on implicit or explicit assumptions about data distribution, as seen in methods like Graph-Q-SAT or MapleSAT-BMM. However, in SAT-based cryptanalysis, different ciphers (e.g., SIMON, MD4), round numbers, or I/O constraints lead to substantially different CNF/ANF structures. Even within the same setting, distribution shifts may arise due to limited training coverage or mismatches with model assumptions, making it difficult for models trained on one problem family to generalize to others.
>
> - **Complexity Sensitivity and Variable Explosion**
>
>     ML methods demonstrating strong performance on simpler instances (e.g., reduced rounds or shorter key lengths) often face significant challenges when applied to larger-scale, more complex problems. This is primarily due to the non-linear growth in the number of variables and clauses as problem size increases, which can cause previously learned representations—such as embeddings or heuristics—to lose relevance or become unstable. Moreover, complex instances (e.g., with longer rounds or larger key lengths) inherently differ in difficulty and structure from simpler ones, meaning that approaches effective on small-scale problems may not translate well to more challenging scenarios. This fundamental gap underscores the need for more scalable and adaptive ML strategies tailored to the complexities of larger cryptographic problems.
>
> Second, to address these challenges, we would like to deliver three solution strategies that may mitigate the generalization problem and we will add these to the final version of the paper:
>
> 1. **Internal ML-based SAT solver with adaptive capability.**
>
>     As in Section 4.2 and Table 2, solvers like MapleCOMSPS-BMM and MapleSAT-BMM embed ML heuristics directly, avoiding costly external calls. This improves scalability, maintaining classical solver runtimes on simple tasks while boosting performance on hard cryptographic instances, thus reducing overhead but keeping adaptability.
>
> 2. **External ML-based initialization for solvers**
>
>     Solvers like WDsat and BASIN use external ML models only once at the start to guide variable selection or structure preprocessing, balancing efficiency and generalization:
>
> - **WDsat** applies ML to solve a Minimum Vertex Cover instance derived from ANF variable interactions (following [Trimoska et al., arXiv:2001.11229]), which reduces formula size via Gaussian elimination. The process is general across datasets, adds negligible overhead to small instances, and improves scalability.
>
> - **BASIN(ours)** predicts initial key assignments from plaintext–ciphertext pairs via a neural network and uses bitwise enumeration applying SIMON round functions to verify candidates. Despite imperfect predictions, enumeration guarantees correctness and bounded runtime. Table 4 shows BASIN’s consistent performance across SIMON rounds with low overhead and strong generalization.
>
> 3. **Domain-specific Optimization in Practice**
>
>    With some domain-specific training data—even limited—ML-based hyperparameter tuning can notably improve performance without runtime overhead. This targeted approach boosts generalization on task-relevant data, as shown in our Q2 response.
>
> ---
>
> **Q3 About the resource requirements**
>
> **A3** Thank you for your comment. The 192-core CPUs and Ascend AI processors mentioned in Appendix A.1 simply describe **the computational resources we used in our experiments, not the resource requirements for running the solvers**. The results are reproducible on different computational resources. In addition, we will update the documentation with clearer instructions for running the code and a more detailed description of the dataset structure to further facilitate reproducibility.
>
> ---
>
> **Q4 About the dense text of the ANF representations and BASIN’s details**
>
> **A4** Thank you for your comment. We will follow your suggestion to revise this in the final version of the paper.
>
> Besides, we will provide a further explanation for you in two aspects:
>
> 1. **For the ANF representations**, we provide detailed descriptions of how common cryptographic operations are expressed in ANF form in Appendix B.1. Additionally, an example of the ANF formula for a SIMON round function is included in Appendix B.2.
>
>     As Appendix B.1 shows, **Algebraic Normal Form(ANF) consist of Boolean equations, representing the conjunction of logical formulas in SAT instances.** Each variable is referred to as a (vanilla) literal. In a Boolean equation, the right side is always 0 and the Boolean function on the left side, formed by the XOR connection of monomials, is called a clause. Each monomial is either a constant term 1, a (vanilla) literal, or a product of variables(literals).
>
> 4. **For the BASIN’s details**, it stands for Bitwise Arithmetic Solver with Initialization by Neural Network and its solving process for SIMON-related datasets is as follows:
>     - (1) **Use A neural network** processes ANF representations derived from input plaintext–ciphertext pairs and **predicts likely initial assignments for key variables**.
>     - (2) **Reads the input in the form of plaintext–ciphertext pairs**, and use the predicted initial assignments as **arguments**. .
>     - (3) **Performs bitwise enumeration** based on the input plaintext-ciphertext pairs and the initial values.
>
>     In essence, it sequentially explores candidate key values from the initial values using bitwise operations: for each candidate, it applies the known SIMON round functions to compute partial ciphertext bits and checks their consistency with the given ciphertext.
>
> ---
>
> **Q5 About the methods over existing works**
>
> **A5** Thank you for your thoughtful and encouraging review,  and we appreciate your recognition of our contributions.
>
> Regarding your comment on the incremental nature of some methods, we agree that our focus in this paper is more on systematic analysis and benchmarking than proposing radically new solver architectures. That said, we aim to address generalization and efficiency challenges through the three solution strategies in Q2, which may be helpful for future work, and we will add these to the final version of the paper.

---

> > ### Author Response · Authors · 2025-08-03
> >
> > Thank you very much for your great efforts in reviewing this paper.

---

> > > ### Comment · Reviewer_ytKu · 2025-08-08
> > > **Response**
> > >
> > > I appreciate your responses. I have no additional concerns and would like to maintain my positive evaluation.

---

> > ### Author Response · Authors · 2025-08-05
> >
> > Thank you very much for your thoughtful comment. We sincerely appreciate your kind words about our work and your careful review.
> >
> > Please don't hesitate to share any further comments or questions.
> >
> > ---
> >
> > > Thank you for your reply, you have solved my question.
> >
> > Also, just a small note: we noticed that your latest comment appears under another reviewer’s thread—just in case that was unintentional.

---

### Official Review · Reviewer_nzcS · 2025-07-09

**Rating:** 4
**Confidence:** 2

**Summary:**

The paper introduces a benchmark for assessing machine learning–based solvers in cryptanalysis and SAT solvers. Particularly, a collection of datasets is introduced that can be used to benchmark various cryptographic algorithms with multiple representations. In addition,  the authors propose a new solver (BASIN) for plaintext-ciphertext input bitstrings. Then, an empirical analysis is provided to compare various ML-based solvers. Based on these experiments, the authors concluded that ML models struggle to generalize across different cryptographic algorithms and instance sizes. But, they showed that ML-driven optimization strategies notably improve solver efficiency on cryptographic SAT instances.

**Dataset Code Accessibility:**

Yes

**Ethical Considerations:**

No, there are no or only very minor ethics concerns

**Final Justification:**

The paper is a nice read on this topic and provides Valuable insights with numerical results on the applicability of ML-driven techniques. The paper benchmarks existing methods using five chosen data sets. I asked about the reason why these data sets have been chosen. The rebuttal discussed the properties of each dataset and how they might be relevant to the problem studied in this paper. However, given that there are other relevant datasets, I'm still not entirely convinced about the significance of choosing these datasets.

**Limitations Weaknesses:**

I am not an expert in this area, but the paper does not clearly convey the reliability, significance, or difficulty of the chosen benchmarking datasets. It remains unclear why these five specific datasets were selected over others. What characteristics make them particularly relevant or distinct compared to existing benchmarks? Furthermore, the paper would benefit from a clearer explanation of what challenges each dataset poses to the solvers and what specific aspects of the problem they are intended to test.

**Strengths Contributions:**

The comprehensive benchmarking framework the paper introduces seems necessary and comprehensive. The empirical studies in the paper are interesting and highlight some of the challenges of ML-driven solvers.

---

> ### Author Rebuttal · Authors · 2025-07-30
>
> Thank you for your comment. We have revised and expanded the documentation for our benchmark to better support reproducibility. In this rebuttal, we have also scaled up the evaluation and included additional experimental results on larger datasets to further demonstrate the reliability of our benchmark. **The dataset documentation has been shown in our Dataset URL.**
>
> Due to rebuttal submission rules, the fully integrated documentation, along with the extended dataset and reorganized code, will be released in the final version, and we would be happy to address any further questions regarding the current dataset or the code.
>
> ---
>
> **Q Why are the five specific datasets in the paper selected over others?**
>
> **A** Thank you for this comment. We will clarify the motivation and characteristics of the selected datasets in the revision.
>
> We chose Simon, MD4, SHA-1, SHA-256, and a ARX-based cipher (“Cipher”) based on their relevance to cryptographic SAT solving and the diverse challenges they pose:
>
> - **Relevance and representativeness**.
> These primitives cover different cryptographic design paradigms:
>
>     - **Simon** (lightweight block cipher with bitwise AND/XOR),
>
>     - **MD4** and **SHA-1** (classic Merkle–Damgård hash functions with dense XOR constraints),
>
>     - **SHA-256** (modern hash with modular additions),
>
>     - **Cipher**(Addition–Rotation–XOR construction).
>
>     They complement generic SAT benchmarks by focusing on cryptanalysis-oriented instances rarely covered in standard suites.
>
> - **Range of difficulty and structure**.
>     Multiple-round variants of Simon and Cipher, configured with different key and plaintext sizes, demonstrate how solver performance scales with increasing algebraic degree.
>
>     Besides, Cipher, designed to mimic ARX cipher decryption with unknown round functions, shows the challenges of reasoning under partially hidden structures.
>
>     In additional, MD4/SHA-1 emphasize XOR-heavy reasoning, and SHA-256 introduces high clause density and modular addition constraints, making it challenging.
>
> - **Distinct challenges to SAT solvers**.
>
>     Each dataset targets different weaknesses in SAT solvers:
>
>     - **Simon**: tests solvers on highly regular bitwise equations with low algebraic degree, which are easier for XOR-aware reasoning but still hard beyond a certain number of rounds.
>
>     - **MD4/SHA-1**: introduce complex XOR-linear relations, stressing clause learning and conflict-driven search.
>
>     - **SHA-256**: adds modular addition constraints that create long-range variable dependencies, which are difficult for CDCL solvers.
>
>     - **Cipher**: evaluates solvers on ARX-dominant constraints without any known algebraic backdoor, providing a “black-box” hardness closer to real-world ciphers.
>
> Overall, compared to existing benchmarks (e.g., SAT Competition suites), these datasets are more specialized toward cryptanalysis-oriented SAT instances, including highly challenging cases where a single instance can require over 5000 seconds to solve. They cover both well-studied cryptographic structures (Simon, MD4, SHA-1, SHA-256) and unknown-round functions (Cipher), thereby better reflecting practical cryptographic challenges.

---

> > ### Author Response · Authors · 2025-08-06
> >
> > Dear Reviewer nzcS,
> >
> > Thank you very much for taking the time to review our submission — We sincerely appreciate your kind words about our work and your careful review.
> >
> > We noticed that an ACK was submitted without additional comments, and we just wanted to kindly follow up to see if there might be any further questions or concerns we could address.
> >
> > Please feel free to let us know if there’s anything further you'd like to discuss. We're happy to provide any clarifications that might assist your evaluation.
> >
> > Thank you again for your time and consideration.
> >
> > Best regards,
> >
> > Authors

---

### Official Review · Reviewer_cTdE · 2025-07-10

**Rating:** 5
**Confidence:** 1

**Summary:**

This paper presents SAT4CryptoBench, a benchmark for testing ML SAT solvers for cryptography. The benchmarks supports solvers with different levels of ML integration, as well as various ways to represent the input problem. The dataset for the benchmarks is generated by converting various cryptographic problems into CDF, ANF and plaintext-cyphertext bitstrings. It is observed that no one existing method is better than others, ML models have issues with generalization and high computational cost, and that ANF-based methods are more powerful than those based on CDF.
On top of that, a new bitwise solver BASIN is proposed, which outperforms both CDF- and ANF-based solvers in certain scenarios.

**Additional Feedback:**

I believe that ANF stands for "algebraic normal form", and not "arithmetic" as you use.
Could you please fix this, or elaborate how it is different?

**Dataset Code Accessibility:**

Partly

**Dataset Code Comments:**

The code and the dataset are not well-documented as described above.

**Ethical Considerations:**

No, there are no or only very minor ethics concerns

**Final Justification:**

From the point of view of technical quality, assuming that the authors will indeed add the promised code to the repository I find the paper to be good enough to publish.

On the other hand, from the perspective of significance or originality I can't judge the quality of the paper since I am not familiar with the field. I would defer it to the judgment of more confident reviewers.

**Limitations Weaknesses:**

While the code is provided, reproducibility suffers from instructions on how to run it
being vaguely written, with part of them being and in Chinese.
Please make it more accessible, i.e. translate all descriptions and comments to English,
add a requirements for the code in the README and exact commands to run to reproduce the results.

The dataset is published, but the documentation of it is too poor.
I would advise to add more information for it to be an independent article. That is, add a description about what the dataset if for and how to use it.

**Strengths Contributions:**

The work is fairly significant, as it proposed the first benchmark that compares different types of solvers for different representation across various cryptographic problems.
The paper overall is fairly well written.

---

> ### Author Rebuttal · Authors · 2025-07-31
>
> We greatly appreciate your time and efforts in this review, and we would be happy to address any further questions regarding the current dataset or the code.
>
> ---
>
> **Q1: Request for more accessible code and dataset documentation**
>
> **A1** Thank you for your valuable feedback. We apologize for any inconvenience caused by the limited initial documentation. In response, we have taken several steps to improve the clarity and accessibility of both the code and the dataset:
>
> - **Code and dependencies**: While the rebuttal rules prevent us from modifying the code repository during this stage, we would like to clarify that the current repository already includes the necessary dependencies for running the external ML-based modules—these are specified in the requirements.txt files located within the corresponding subfolders/zips.
>
> - **Documentation**: We have revised the benchmark documentation to enhance reproducibility. **Documentation is available at our Dataset URL.** Besides, we have supplemented the codebase with a more detailed English README for the BASIN to enhance clarity and reproducibility. Due to rebuttal constraints, we present this README as plain text within the rebuttal (see below).
>
> - **Future improvements**: Due to rebuttal constraints, a more comprehensive reorganization of the code and an expanded version of the dataset will be included in the final version, which will further improve modularity and ease of reproduction.
>
> We have also expanded our evaluation with additional experiments on larger datasets to better demonstrate the robustness and reliability of our benchmark, and we would be happy to address any further questions regarding the current dataset or the code.
>
> ---
>
> Updated README for BASIN:
>
> ---
> # 🔐 Neural Initialization + SAT Solver Pipeline for SIMON-12/32/64
>
> This project provides a two-stage automated script that integrates a neural key prediction model with a SIMON cryptanalysis solver, enabling efficient initialization and solving of SAT instances represented in ANF format.
>
> ---
>
> ## 📁 Folder structure
>
> ```plaintext
> ├── compute_time.py              # Main execution script
> ├── prediction.py                # Neural network predictor script
> ├── simon_anf                    # Precompiled SIMON ANF solver binary
> ├── simon-12-32-64-final         # 'anf' folder stores ANF-format SAT instances
>                                  # 'cnf' folder stores CNF-format SAT instances
>                                  # 'plain_cipher' folder contains plaintext-ciphertext pairs in TXT format
>
> ```
>
> ---
>
> ## ⚙️ Script Functionality Overview
>
> `compute_time.py` performs two main stages:
>
> ### 🧠 Stage 1: Key Bit Initialization via Neural Prediction
>
> A neural model (here we use CryptoANFNet) predicts a 32-bit key string to initialize each SAT instance for accelerated solving.
>
> The prediction is invoked via the following command:
>
> ```python
> predict_cmd = (
>     f"python prediction.py assignment {cnf_path} {CHECKPOINT_PATH} "
>     f"--graph anf --seed 123 --model neurosat --test_splits sat"
> )
> ```
>
> - `assignment`: Specifies the task type as “variable assignment”
> - `{cnf_path}`: Path to the SAT instance in ANF format
> - `{CHECKPOINT_PATH}`: Path to the trained neural model checkpoint
> - `--graph anf`: Use ANF-based graph construction
> - `--model neurosat`: Specifies the neural model (Here, we implemented **a new version of CryptoANFNet based on NeuroSAT**, and thus refer to it as NeuroSAT throughout for simplicity.)
> - `--test_splits sat`: Use sat as the test split
>
> The output will contain a 32-bit predicted key string.
>
> ### ⚙️ Stage 2: Solving with the Predicted Key
>
> The `simon_anf` solver is executed as follows:
>
> ```bash
> ./simon_anf <path_to_txt_file> <predicted_key_string>
> ```
>
> - `<path_to_txt_file>`: Path to the `.txt` file containing plaintext-ciphertext pairs
> - `<predicted_key_string>`：32-bit predicted key string from Stage 1
>
> This binary will solve the ANF instance using the provided plaintext-ciphertext pair and initialization key.
>
> ---
>
> ## 🔧 Configuration
>
> ```python
> TXT_FOLDER = "<path_to_plain_cipher_folder>"
> CNF_FOLDER = "<path_to_ANF_instances>"
> CHECKPOINT_PATH = "<path_to_trained_model_checkpoint>"
> ```
>
> Update the above paths to match your local environment.
> Ensure that files in the plain_cipher and anf folders follow a consistent naming pattern (e.g., `0.txt` and `0.cnf`).
>
> ---
>
> ## 📌 Input Format
>
> - `*.txt`： Contains plaintext and ciphertext (used during solving).
> - `*.cnf`： ANF-format SAT instance, with the same base filename as the corresponding `.txt` file (used for prediction)
>
> ---
>
> ## 🚀 How to Run
>
> Make sure the CNF files, TXT files, and trained model checkpoint are ready. Then run:
>
> ```bash
> python compute_time.py
> ```
>
> ---
> ## 📝 Output
>
> - The prediction stage will output a 32-bit key prediction for each instance
> - The solver stage will report the solving time for each instance
> - A final average solving time will be reported at the end
>
> ---
> ---
>
> **Q2: About “algebraic normal form”**
>
> **A2** Thank you for your comment. We will fix this in the final version of the paper.

---

> > ### Comment · Reviewer_cTdE · 2025-08-01
> >
> > Thank you for your response.
> >
> > With the provided README file it is still unclear to me how to reproduce your results or run your code.
> > Particularly, you mentioned need that "trained model checkpoint are ready" before running the code, it is not mentioned how to obtain them. Did you provide them somewhere in the repository?
> >
> > More broadly, what I expect from the code is that you provide a sequence of command line calls in a README or in a separate shell file that, when executed on a clean system, will generate the same tables that you have in your paper. Ideally you should also provide a Docker image that does it, although I understand that this is not allowed during the rebuttal.
> > Nevertheless, I would still appreciate it if you could provide the aforementioned shell file as a plain text.

---

> > ### Author Response · Authors · 2025-08-02
> >
> > **Q1: About how to run BASIN?**
> >
> > Thank you for your comment. We have provided an example checkpoint file at `BASIN/ckpt.pt`. To reproduce our results, you can download the code and run the experiment by replacing the original python `BASIN/compute_time.py` command with the following, or alternatively, modify the file paths in `compute_time.py` as you prefer.
> >
> > (Please make sure to set `BASIN/simon_anf` as executable, for example by running `chmod 777 simon_anf`.)
> >
> > ```python
> > import os
> > import subprocess
> > import time
> >
> > TXT_FOLDER = "./simon-12-32-64-final/plain_cipher/test"
> > CNF_FOLDER = "./simon-12-32-64-final/anf/test/sat"
> > CHECKPOINT_PATH = "./ckpt.pt"
> >
> > def get_cnf_path(txt_path):
> >     filename = os.path.basename(txt_path).replace(".txt", ".cnf")
> >     return os.path.join(CNF_FOLDER, filename)
> >
> > def parse_prediction_output(output_str):
> >     for line in output_str.strip().splitlines():
> >         line = line.strip()
> >         if len(line) == 32 and all(c in '01' for c in line):
> >             return line
> >     return None
> >
> > def predict_all_initializations(txt_files):
> >     init_dict = {}
> >     print("🧠 Generating initialization bitstrings using neural network...\n")
> >
> >     for f in txt_files:
> >         txt_path = os.path.join(TXT_FOLDER, f)
> >         cnf_path = get_cnf_path(txt_path)
> >
> >         if not os.path.exists(cnf_path):
> >             print(f"❌ Missing CNF file for {f}")
> >             continue
> >
> >         predict_cmd = (
> >             f"python prediction.py assignment {cnf_path} {CHECKPOINT_PATH} "
> >             f"--graph anf --seed 123 --model neurosat --test_splits sat"
> >         )
> >
> >         prediction_proc = subprocess.run(predict_cmd, shell=True, capture_output=True, text=True)
> >         if prediction_proc.returncode != 0:
> >             print(f"❌ Prediction failed for {f}:\n{prediction_proc.stderr}")
> >             continue
> >
> >         bitstring = parse_prediction_output(prediction_proc.stdout)
> >         if bitstring is None:
> >             print(f"❌ Could not extract valid bitstring from prediction for {f}")
> >             continue
> >
> >         init_dict[f] = bitstring
> >
> >     return init_dict
> >
> > def solve_with_initializations(txt_files, init_dict, init_time):
> >     total_times = []
> >     print("⚙️  Solving with predicted initializations...\n")
> >
> >     for f in txt_files:
> >         if f not in init_dict:
> >             print(f"⚠️  Skipping {f}: no init bitstring found.")
> >             continue
> >
> >         txt_path = os.path.join(TXT_FOLDER, f)
> >         bitstring = init_dict[f]
> >
> >         start = time.time()
> >         try:
> >             subprocess.run(["./simon_anf", txt_path, bitstring], check=True, stdout=subprocess.PIPE, stderr=subprocess.PIPE)
> >             end = time.time()
> >             total_times.append(end - start)
> >             print(f"⏱️  {f}: {end - start:.4f} seconds")
> >         except subprocess.CalledProcessError as e:
> >             print(f"❌ SIMON failed on {f}: {e}")
> >
> >     if total_times:
> >         avg_time = (sum(total_times) + init_time) / len(total_times)
> >         print(f"\n📊 Average solving time (excluding prediction): {avg_time:.4f} seconds over {len(total_times)} files.")
> >     else:
> >         print("❌ No successful solves.")
> >
> > def main():
> >     if not os.path.isdir(TXT_FOLDER):
> >         print(f"❌ TXT folder not found: {TXT_FOLDER}")
> >         return
> >
> >     txt_files = sorted(f for f in os.listdir(TXT_FOLDER) if f.endswith(".txt"))
> >     if not txt_files:
> >         print("⚠️ No .txt files found.")
> >         return
> >
> >     pre_start = time.time()
> >     init_dict = predict_all_initializations(txt_files)
> >     pre_end = time.time()
> >     init_time = pre_end - pre_start
> >     print(f"process_time: {init_time}")
> >
> >     solve_with_initializations(txt_files, init_dict, init_time)
> >
> > if __name__ == "__main__":
> >     main()
> > ```
> >
> > You will obtain output similar to the following:
> >
> > ```bash
> > 🧠 Generating initialization bitstrings using neural network...
> >
> > process_time: 637.0434248447418
> > ⚙️  Solving with predicted initializations...
> >
> > ⏱️  0.txt: 33.8703 seconds
> > ⏱️  1.txt: 34.4434 seconds
> > ...
> >
> > 📊 Average solving time (excluding prediction): 43.5337 seconds over 100 files.
> > ```
> >
> > Note that instances in the TXT_FOLDER follow the format below (using 0.txt as an example):
> > ```
> > plaintext:[1, ..., 0, 1]
> > ciphertext:[1, ..., 1, 0]
> > key:[1, 0, ..., 0]
> > ```
> > Each instance provides a plaintext, a ciphertext, and the corresponding ground-truth key used for training the neural network. During testing, in fact, the ground-truth key is not used in the solving process (and the `key:[...` line can be removed from the `.txt` file).
> >
> > If you print the output from the subprocess (e.g., using `result = subprocess.run(...)` and `print(result.stdout)`), you will see the key recovered by our solver.
> >
> > ---
> > The required dependencies are listed in `BASIN/requirements.txt`.
> > Besides, if your environment does not yet include `torch-geometric` and `torch-scatter`, please make sure to install them.
> >
> > We greatly appreciate your feedback, and we will similarly reorganize the code to ensure that the experimental results can be easily reproduced when the repository is released.

---

> > > ### Author Response · Authors · 2025-08-02
> > >
> > > **Q2: About command sequence for experiments**
> > >
> > > Thank you for your valuable suggestion, and we sincerely apologize for the inconvenience caused. In this comment, we have now provided the exact commands and code used in our experiments for testing.
> > >
> > > **For Heuristic Enhancement**, we used the following code to evaluate the solvers.
> > > (Please make sure to compile the required solvers as described in the corresponding README files beforehand. For example, to obtain the `execution_path` of maplesat, you can extract `ML-enhanced-Solver/maplesat.zip` and follow the instructions in `ML-enhanced-Solver/maplesat/README.md` to compile `maplesat`.)
> > >
> > > ```python
> > > import os
> > > import shutil
> > > import subprocess
> > > import random
> > > import time
> > > import logging
> > > import concurrent.futures
> > >
> > > EXPERIMENT_NUM = 10
> > > SOLVER_NUM = 10
> > > execution_path = '' ## Change this to your own execution command for your solver
> > > ## Such as MapleSAT: ./maplesat -no-luby -rinc=1.5 -phase-saving=0 -rnd-freq=0.02
> > >
> > > solver_options = ['MaplePainless', 'CDCL-Crypto', 'Maplesat', 'glucose-4.1-bmm', 'maplecomsps_lrb_vsids_18-bmm', 'maplelcmdistchronobt-bmm',
> > >                 'maplesat-bmm', 'glucose-4.1', 'maplecomsps_lrb_vsids_18', 'maplelcmdistchronobt', 'cadical']
> > > data_options = ['8rounds-sat', '9rounds-sat', '10rounds-sat', '11rounds-sat', '12rounds-sat',
> > >                 'simon-10-32-64-final-sat', 'simon-11-32-64-final-sat', 'simon-12-32-64-final-sat',
> > >                 'md4-20rounds', 'sha1-21rounds', 'sha256-18rounds']
> > >
> > > data_path_options = [
> > >     '8rounds/sat', '9rounds/sat', '10rounds/sat', '11rounds/sat', '12rounds/sat',
> > >     'simon/simon-10-32-64-final/sat', 'simon/simon-11-32-64-final/sat', 'simon/simon-12-32-64-final/sat',
> > >     'crypto_encoding/md4-20rounds', 'crypto_encoding/sha1-21rounds', 'crypto_encoding/sha256-18rounds'
> > > ]
> > >
> > > data_path = f'../{data_path_options[EXPERIMENT_NUM]}' ## Change this to your own data path
> > > cnf_files = [f for f in os.listdir(data_path) if f.endswith('.cnf')]
> > >
> > > log_path = f'./logs/{solver_options[SOLVER_NUM]}' ## Change this to your own log path
> > > os.makedirs(log_path, exist_ok=True)
> > > logging.basicConfig(
> > >     filename=os.path.join(log_path, f'{data_options[EXPERIMENT_NUM]}.log'), ## Change this to your own log name
> > >     level=logging.INFO,
> > >     format='%(asctime)s - %(levelname)s - %(message)s'
> > > )
> > >
> > > def run_single(execution_path, cnf_path):
> > >     command = f'{execution_path} {cnf_path}'
> > >     start_time = time.time()
> > >     try:
> > >         result = subprocess.run(command, shell=True, capture_output=True, text=True, timeout=5000)
> > >         end_time = time.time()
> > >         execution_time = end_time - start_time
> > >         log_msg = f'{cnf_path} took {execution_time} seconds'
> > >         logging.info(log_msg)
> > >     except subprocess.TimeoutExpired as e:
> > >         end_time = time.time()
> > >         log_msg = f'{cnf_path} took 5000 seconds'
> > >         logging.info(log_msg)
> > >
> > > if __name__ == '__main__':
> > >     logging.info('Start Running')
> > >     with concurrent.futures.ThreadPoolExecutor(max_workers=5) as executor:
> > >         futures = []
> > >         for cnf_file in cnf_files:
> > >             cnf_path = os.path.join(data_path, cnf_file)
> > >             futures.append(executor.submit(run_single, execution_path, cnf_path))
> > >         concurrent.futures.wait(futures)
> > > ```
> > >
> > > Specially, for GraphQSat, NeuroBack, and neuro-cadical, we manually modified all hardcoded paths in their original scripts (see their folders/zips). Then, we used Huawei Ascend's migration script to convert torch.cuda calls into torch.npu. Finally, we resolved a variety of dependency and compilation issues that arose during this process.
> > >
> > > The Huawei migration script is generally used in the following format:
> > > ```
> > > ${ASCEND_TOOLKIT_HOME}/latest/tools/ms_fmk_transplt/pytorch_gpu2npu.sh \
> > >   -i <path_to_original_script> \
> > >   -o <path_to_output_migrated_script> \
> > >   -v <pytorch_version>
> > > ```
> > > Note that this testing setup may not be easily reproducible on machines without Ascend hardware. In response to your suggestion, we will provide either a Docker image or alternative testing instructions for non-Ascend GPU environments when the repository is released.
> > >
> > > **For Standalone Distinguisher**, please refer to `Standalone Solver/README.md`. We will also include the exact model files used in our experiments when the repository is released.
> > >
> > > **For Hyperparameter Optimization**, the full search procedure can be found in: `Hyperparameter-Optimization/search_optim_params/EasyNAS-dev_evaluater` folder and `Hyperparameter-Optimization/search_optim_params/hebo_optim_kissat.py`
> > >
> > > To reproduce results with optimized hyperparameters (without re-running the search), reuse the script from the **Heuristic Enhancement** experiments and adjust the execution_path accordingly (e.g., `./kissat --restartint=1 --reduceint=1000 --decay=50`). The exact settings are listed in **Table 6 (Appendix)**.
> > >
> > > We hope this clarifies our experimental setup. We greatly appreciate your feedback and will restructure the code and provide clearer instructions to ensure reproducibility upon public release.

---

> > > > ### Comment · Reviewer_cTdE · 2025-08-05
> > > >
> > > > Thank you for updating the code.

---

> > ### Author Response · Authors · 2025-08-05
> >
> > Thank you very much for your valuable comments and for the time and effort you’ve devoted to reviewing our paper.
> >
> > We sincerely appreciate your acknowledgment of our work and your detailed suggestions on reproducibility.
> >
> > Please don’t hesitate to share any further thoughts or questions.
> >
> > ---
> >
> > In addition, we will make every effort to ensure that the final released version of our code and dataset is easy to use, to support reproducing our results and facilitating future research. Once available, please feel free to raise any issues or suggestions regarding their usage—we would be glad to assist.

---

### Official Review · Reviewer_YNLq · 2025-07-18

**Rating:** 4
**Confidence:** 3

**Summary:**

This paper presents SAT4CryptoBench, a benchmark for evaluating ML-based SAT solvers on cryptographic problems. It includes datasets in ANF, CNF, and supports three ML integration levels: distinguishers, heuristics, and hyperparameter tuning. The authors also introduce BASIN, a bitwise solver using plaintext-ciphertext inputs, which performs well on complex tasks. Experiments show how input format and integration strategy impact solver performance.

**Dataset Code Accessibility:**

Yes

**Ethical Considerations:**

No, there are no or only very minor ethics concerns

**Final Justification:**

The authors have addressed the concerns, and I hope all revisions will be incorporated into the final version.

**Limitations Weaknesses:**

- The work is short of proposing or validating new solver architectures or optimization strategies that address key limitations of ML-based solvers, including poor generalization across algorithms and instance sizes, as well as high computational overhead on simpler tasks.

- Additionally, the ANF and CNF datasets are relatively small, with only 210 instances per algorithm (10 for training and 200 for testing), limiting the depth of analysis.

- The paper also lacks statistical significance reporting (e.g., variance, confidence intervals), making it hard to judge the robustness of the results.

- Finally, it misses direct comparisons with both traditional and existing ML-based solvers, which would help contextualize the benchmark’s utility.

**Strengths Contributions:**

The paper fills a key gap by introducing SAT4CryptoBench, a standardized and reproducible benchmark for ML-based SAT solvers in cryptographic settings.

- It supports three ML integration paradigms—standalone, heuristic-enhanced, and hyperparameter-tuned—offering a comprehensive performance view.

- Datasets in ANF, CNF, and bitstring formats allow for controlled input representation analysis.

- The writing is clear and the structure well-organized.

---

> ### Author Rebuttal · Authors · 2025-07-30
>
> Thanks for your comment. We have revised the benchmark documentation to enhance reproducibility. In this rebuttal, we also scaled up evaluations and added results on larger datasets to further validate the benchmark. **Documentation is available at our Dataset URL**
>
> Due to rebuttal constraints, full documentation, extended datasets, and reorganized code will be released in the final version. We welcome any further questions about the current dataset or code.
>
> ---
> **Q1 About proposing new solver architectures or optimizations for ML-based solver limitations**
>
> **A1** Thanks. While we do not propose a new architecture, our main contribution is SAT4CryptoBench—a unified benchmark for systematic evaluation of ML-based SAT solvers in cryptanalysis. Our goals are:
>
> - **Standardized datasets** across formats, capturing cryptographic structure
>
> - **Reproducible evaluation** of ML integration strategies
>
> - **Exposure of key limitations**, such as weak generalization and overhead on simple tasks
>
> Our benchmark makes such issues visible and reproducible, motivating future improvements.
>
> Here, we propose three practical strategies to address these issues (to be added in the final version):
>
> 1. **Internal ML-based SAT solver with adaptive capability.**
>
>     As in Section 4.2 and Table 2, solvers like MapleCOMSPS-BMM and MapleSAT-BMM embed ML heuristics directly, avoiding costly external calls. This improves scalability, maintaining classical solver runtimes on simple tasks while boosting performance on hard cryptographic instances, thus reducing overhead but keeping adaptability.
>
>
> 2. **External ML-based initialization for solvers**
>
>     Solvers like WDsat and BASIN use external ML models only once at the start to guide variable selection or structure preprocessing, balancing efficiency and generalization:
>
> - **WDsat** applies ML to solve a Minimum Vertex Cover instance derived from ANF variable interactions (following [Trimoska et al., arXiv:2001.11229]), which reduces formula size via Gaussian elimination. The process is general across datasets, adds negligible overhead to small instances, and improves scalability.
>
> - **BASIN(ours)** predicts initial key assignments from plaintext–ciphertext pairs via a neural network and uses bitwise enumeration applying SIMON round functions to verify candidates. Despite imperfect predictions, enumeration guarantees correctness and bounded runtime. Table 4 shows BASIN’s consistent performance across SIMON rounds with low overhead and strong generalization.
>
> 3. **Domain-specific Optimization in Practice**
>
>    With some domain-specific training data—even limited—ML-based hyperparameter tuning can notably improve performance without runtime overhead. This targeted approach boosts generalization on task-relevant data, as shown in our Q2 response.
>
> ---
> **Q2 About the scale of the dataset**
>
> **A2** Thanks. We would like to clarify that the 210-instance-per-algorithm setting (10 for training, 200 for testing) is specific to **hyperparameter optimization**.  Additional experimental settings are in **Appendix A**.
>
> For hyperparameter optimization, our aim is to show that even with limited training samples, optimization algorithms can find effective hyperparameters that improve solver performance over other ML-based methods.
>
> In fact, **increasing the search budget leads to better results.** For example, on the **Simon-12-32-64** dataset with Kissat (**100 training, 3000 testing**), HEBO and EasyNAS find hyperparameters that greatly reduce solving time compared to the baseline:
>
> |Solvers|Method|Hyperparameters|Time(s)|
> |----|----|----|----|
> |Kissat|Baseline|default| 760.54 $\pm$ 30.52|
> |Kissat|HEBO|restartint=5164, reduceint=945, decay=1, eliminateinit=1519, rephaseint=4996| 358.76 $\pm$ 11.03|
> |Kissat|EasyNAS|restartint=23, reduceint=1646, decay=1, eliminateinit=111, rephaseint=4838|394.49 $\pm$ 11.46|
>
> (95% Confidence Interval shown)
>
> We have also evaluated on a **test set of 1000** with the **same hyperparameters from Table 3**. Results(below) remain consistent and confirm that hyperparameter optimization significantly improves efficiency. Furthermore, as noted in Q3, ML-enhanced solvers have also been evaluated on extended datasets:
>
> |Solvers|Method|Simon-12-32-64|Cipher-12|MD4|SHA-256|SHA-1|
> |----|----|----|----|----|----|----|
> ||Baseline|818.92 $\pm$ 55.93|974.35 $\pm$ 66.02|0.078 $\pm$ 0.001|88.57 $\pm$ 7.61|68.92 $\pm$ 6.28|
> |Kissat|HEBO|381.46 $\pm$ 20.10|521.42 $\pm$ 25.25|0.059 $\pm$ 0.001|76.79 $\pm$ 7.23|50.33 $\pm$ 6.25|
> ||EasyNAS|595.47 $\pm$ 30.67|583.55 $\pm$ 27.69|0.062 $\pm$ 0.001|78.57 $\pm$ 7.49|53.86 $\pm$ 5.50|
> ||Baseline|3349.52 $\pm$ 115.28|3212.63 $\pm$ 114.04|0.076 $\pm$ 0.001|745.24 $\pm$ 54.18|227.27 $\pm$ 9,59|
> |Cryptominisat|HEBO|3027.48 $\pm$ 88.41|2894.72 $\pm$ 116.02|0.026 $\pm$ 0.001|615.73 $\pm$ 50.08|187.46 $\pm$ 10.21|
> ||EasyNAS|3003.01 $\pm$ 88.62|2845.21 $\pm$ 115.53|0.024 $\pm$ 0.001|553.11 $\pm$ 40.25|185.29 $\pm$ 9.83|
> ||Baseline|440.02 $\pm$ 18.77|381.22 $\pm$ 15.94|1.435 $\pm$ 0.013|111.52 $\pm$ 14.41|75.92 $\pm$ 13.07|
> |MapleSAT|HEBO|378.14 $\pm$ 15.73|304.67 $\pm$ 12.97|0.425 $\pm$ 0.007|87.47 $\pm$ 6.91|60.42 $\pm$ 5.42|
> ||EasyNAS|371.78 $\pm$ 14.48|338.17 $\pm$ 14.89|0.276 $\pm$ 0.005|94.72 $\pm$ 8.31|69.87 $\pm$ 6.86|
>
> ---
> **Q3 About the lack of statistical significance reporting**
>
> **A3** Thanks for the suggestion. We have added confidence intervals to all results and evaluated on a larger **test set (size 1000)** to confirm benchmark reliability. The updated results (below) remain consistent with original reports, supporting the robustness of our conclusions.
>
> |Method|Cipher 8|Cipher 9|Cipher 10|Cipher 11|Cipher 12|MD4|Simon 10-32-64|Simon 11-32-64|Simon 12-32-64|SHA 1|SHA 256|
> |----|----|----|----|----|----|----|----|----|----|----|----|
> |MaplePainless|1.07 $\pm$ 0.01|2.97 $\pm$ 0.27|21.21 $\pm$ 2.21|69.40 $\pm$ 7.19|390.22 $\pm$ 46.50|1.430 $\pm$ 0.035|22.00 $\pm$ 2.20|77.10 $\pm$ 7.49|445.31 $\pm$ 50.86|53.89 $\pm$ 7.05|153.32 $\pm$ 23.17|
> |MapleSat|0.27 $\pm$ 0.03|1.02 $\pm$ 0.12|12.94 $\pm$ 1.29|63.12 $\pm$ 9.06|381.22 $\pm$ 15.94|1.435 $\pm$ 0.013|26.46 $\pm$ 5.13|73.85 $\pm$ 10.2|440.02 $\pm$ 18.77|75.92 $\pm$ 13.07|111.52 $\pm$ 14.41|
> |MapleSat-BMM|0.38 $\pm$ 0.04|1.16 $\pm$ 0.13|12.17 $\pm$ 1.51|51.29 $\pm$ 5.44|390.84 $\pm$ 45.89|0.084 $\pm$ 0.001|14.83 $\pm$ 1.83|41.57 $\pm$ 5.05|460.57 $\pm$ 55.87|41.75 $\pm$ 4.86|120.78 $\pm$ 18.31|
> |MapleSat-Crypto|0.34 $\pm$ 0.02|1.54 $\pm$ 0.10|14.25 $\pm$ 1.07|48.36 $\pm$ 4.73|443.11 $\pm$ 35.67|0.271 $\pm$ 0.018|15.98 $\pm$ 3.77|58.34 $\pm$ 5.35|717.31 $\pm$ 64.83|43.21 $\pm$ 4.55|124.27 $\pm$ 16.24|
> |Glucose-BMM|0.21 $\pm$ 0.02|1.28 $\pm$ 0.12|24.56 $\pm$ 5.80|98.74 $\pm$ 14.31|1050.95 $\pm$ 100.79|0.73 $\pm$ 0.07|27.33 $\pm$ 6.32|90.46 $\pm$ 14.08|1388.51 $\pm$ 180.96|80.96 $\pm$ 14.33|5764.04 $\pm$ 769.49|
> |Glucose*|0.22 $\pm$ 0.02|1.36 $\pm$ 0.16|14.34 $\pm$ 1.55|70.93 $\pm$ 10.91|914.88 $\pm$ 100.60|0.084 $\pm$ 0.001|14.52 $\pm$ 2.02|79.15 $\pm$ 12.66|1434.34 $\pm$ 190.33|53.25 $\pm$ 8.71|2932.32 $\pm$ 421.31|
> |MapleCOMSPS-BMM|0.28 $\pm$ 0.03|1.87 $\pm$ 0.26|35.32 $\pm$ 4.10|126.20 $\pm$ 15.32|1002.57 $\pm$ 142.45|0.095 $\pm$ 0.001|32.33 $\pm$ 4.20|110.60 $\pm$ 13.91|1248.37 $\pm$ 213.52|146.34 $\pm$ 20.30|336.59 $\pm$ 47.52|
> |MapleCOMSPS*|0.22 $\pm$ 0.03|1.97 $\pm$ 0.23|29.85 $\pm$ 3.65|105.94 $\pm$ 12.35|1061.95 $\pm$ 130.22|0.102 $\pm$ 0.001|28.77 $\pm$ 3.42|125.78 $\pm$ 13.28|1467.97 $\pm$ 220.32|194.67 $\pm$ 38.04|377.19 $\pm$ 48.88|
> |MapleLCMDist-BMM|0.73 $\pm$ 0.24|3.72 $\pm$ 0.55|28.66 $\pm$ 2.65|75.55 $\pm$ 7.54|368.26 $\pm$ 45.72|0.084 $\pm$ 0.001|24.10 $\pm$ 2.70|63.65 $\pm$ 7.44|382.07 $\pm$ 43.16|51.76 $\pm$ 5.12|226.01 $\pm$ 36.77|
> |MapleLCMDist*|0.11 $\pm$ 0.01|1.10 $\pm$ 0.12|21.49 $\pm$ 2.48|67.88 $\pm$ 6.33|382.92 $\pm$ 40.66|0.089 $\pm$ 0.001|24.41 $\pm$ 2.87|69.08 $\pm$ 6.52|389.66 $\pm$ 44.37|38.40 $\pm$ 5.90|152.86 $\pm$ 21.42|
> |Kissat(NeuroBack)|10.77 $\pm$ 1.59|27.48 $\pm$ 3.22|396.57 $\pm$ 63.75|1536.74 $\pm$ 238.28|2184.22 $\pm$ 271.85|13.539 $\pm$ 0.814|433.77 $\pm$ 61.78|1728.98 $\pm$ 263.36|3783.78 $\pm$ 270.93|651.74 $\pm$ 65.59|803.39 $\pm$ 109.48|
> |Kissat*|0.12 $\pm$ 0.02|0.63 $\pm$ 0.07|12.14 $\pm$ 1.34|63.25 $\pm$ 4.56|974.35 $\pm$ 66.02|0.078 $\pm$ 0.001|11.28 $\pm$ 1.08|73.24 $\pm$ 5.10|818.92 $\pm$ 55.93|68.92 $\pm$ 6.28|88.57 $\pm$ 7.61|
> |Neuro-Cadical|0.95 $\pm$ 0.02|1.61 $\pm$ 0.07|20.68 $\pm$ 2.60|174.41 $\pm$ 22.77|1931.95 $\pm$ 301.49|1.114 $\pm$ 0.020|20.68 $\pm$ 2.85|173.05 $\pm$ 33.01|1910.22 $\pm$ 272.73|402.67 $\pm$ 92.02|433.70 $\pm$ 129.67|
> |Cadical*|0.90 $\pm$ 0.02|1.53 $\pm$ 0.08|22.76 $\pm$ 3.00|181.00 $\pm$ 25.32|1976.53 $\pm$ 293.76|0.864 $\pm$ 0.015|16.07 $\pm$ 2.16|173.71 $\pm$ 25.21|1874.66 $\pm$ 320.17|464.05 $\pm$ 74.14|625.49 $\pm$ 180.72|
> |Minisat(Graph-Q-Sat)|33.96 $\pm$ 2.38|36.78 $\pm$ 2.11|80.53 $\pm$ 5.77|430.15 $\pm$ 43.52|2437.76 $\pm$ 286.59|0.083 $\pm$ 0.001|70.77 $\pm$ 5.25|183.71 $\pm$ 19.39|1737.40 $\pm$ 195.71|2992.97 $\pm$ 307.96|15020.32 $\pm$ 7682.01|
> |Minisat*|0.28 $\pm$ 0.05|1.31 $\pm$ 0.22|17.10 $\pm$ 2.02|124.96 $\pm$ 12.93|1503.46 $\pm$ 167.94|0.259 $\pm$ 0.012|15.62 $\pm$ 2.21|60.73 $\pm$ 6.05|1787.83 $\pm$ 181.61|143.69 $\pm$ 35.86|443.32 $\pm$ 22.89|
>
> (95% Confidence Interval shown)
>
> ---
> **Q4 About direct comparisons with both traditional and existing ML-based solvers**
>
> A4 Thanks. We would like to clarify that there exists a direct comparison in **Table 2** of the paper, where results from traditional solvers are marked by *, and a more comprehensive comparison—including different input formats, datasets, traditional baselines (denoted as base) —is summarized in **Table 7 in the Appendix**. We appreciate your suggestion and will make sure to emphasize this more clearly in the final version.
>
> Additionally, we have conducted experiments on a larger test set (1000 instances). Results remain consistent with original findings, reinforcing the validity of these comparisons (see responses to Q2 \& Q3).

---

> > ### Comment · Reviewer_YNLq · 2025-08-07
> >
> > Thank you for addressing most of the concerns and providing additional experiments. In the Table for Q3, the Cipher12 and SHA 256 methods exhibit high variance — could you provide further explanation for this observation?

---

> > > ### Author Response · Authors · 2025-08-07
> > >
> > > **Q the Cipher12 and SHA 256 methods exhibit high variance**
> > >
> > > **A** Thank you for your feedback. We think your concern refers to the observed high variance in runtime for certain methods on the Cipher12 and SHA-256 datasets. To clarify this observation, we offer the following analysis from several perspectives:
> > >
> > > 1. **Dataset Complexity and Structural Properties**
> > >
> > > Cipher12 involves significantly more encryption rounds than other ciphers in our benchmark, leading to much more complex CNF encodings. Similarly, SHA-256 adds modular addition constraints that create long-range variable dependencies. These structural features lead to highly entangled SAT instances with large search spaces, making solver performance particularly **sensitive to the initial decision path**.
> > >
> > > ---
> > >
> > > 2. **Solver Dynamics and Restart Mechanisms**
> > >
> > > CDCL-based heuristic SAT solvers, whether traditional or enhanced with learnable modules, employ restart and initialization strategies. When the initial search path is close to a solution, it can converge quickly with minimal restarts. Conversely, when the initial direction is far from the solution, the solver may require numerous restarts to adjust its trajectory—each restart being computationally expensive. For challenging datasets like Cipher12 and SHA-256, the variance in runtime arises **mainly from these diverging search trajectories and the high cost of restarts**.
> > >
> > > ---
> > >
> > > 3. **Solvers with Neural Components**
> > >
> > > Solvers incorporating neural components—such as learned branching heuristics used in Graph-Q-SAT—tend to exhibit even greater runtime variance. **The stochastic nature of neural guidance can lead to widely different decision paths across runs**, especially on structurally complex or noisy instances outside the training distribution.  This explains the larger variance observed for such solvers on Cipher12 and SHA-256.
> > >
> > > ---
> > >
> > > 4. **General Observations Across Datasets**
> > >
> > > Importantly, this phenomenon is not unique to Cipher12 and SHA-256. Similar levels of runtime variance have been observed on other comparably difficult datasets, such as Simon 12-32-64. Even on relatively simpler benchmarks like Cipher11 and Simon 11-32-64, the 95% confidence intervals for solver runtimes still span approximately 10% of the average solving time. This suggests that **the phenomenon is a general trait of hard cryptographic SAT instances, not an anomaly of specific solvers or datasets**.
> > >
> > > ---
> > >
> > > We hope these clarifications adequately address your concerns and explain the runtime variance in a broader perspective.

---

> > > > ### Comment · Reviewer_YNLq · 2025-08-07
> > > >
> > > > Thanks for your clarification, I appreciate it.

---

> > > > > ### Author Response · Authors · 2025-08-08
> > > > >
> > > > > Thank you very much for your thoughtful feedback. We sincerely appreciate your acknowledgment of our work and your care in reviewing it. We would be glad to address any additional comments or questions you may have.
> > > > >
> > > > > Please feel free to let us know if there’s anything further you'd like to discuss.

---

### Note · Authors · 2025-08-12

We sincerely appreciate the reviewers’ great efforts and suggestions. Overall, our work was deemed “well-organized/written” (`YNLq, cTdE, n3Jt`), “significant/valuable” (`cTdE, n3Jt`), with a “comprehensive” and “thorough” experiment design (`nzcS, ytKu, n3Jt`). The results were described as “convincing,” “reproducible,” and “interesting” (`nzcS, ytKu, n3Jt`).

The main concerns involve new solver architectures or optimizations for ML-based solver limitations, dataset scale, and the documentation. Here, we restate our contributions and address these points.

---

Contributions

- **Standardized datasets** across formats, capturing cryptographic structure

- **Reproducible evaluation** of ML integration strategies

- **Key finding** such as weak generalization, overhead on simple tasks, and the effectiveness of direct input representations

Our benchmark exposes these findings clearly, encouraging further research.

---

Discussion

1. **New Solver architectures or optimizations for ML-based solvers**(`YNLq,ytKu,n3Jt`)

    Our analysis suggests several strategies:

- **Internal ML integration** (MapleCOMSPS-BMM, MapleSAT-BMM): embeds ML heuristics to avoid external calls, preserving classical runtimes on easy tasks and improving performance on hard cryptographic ones.

- **External ML initialization** (WDsat, BASIN): applies ML once at startup.

    - **WDsat**: solves Minimum Vertex Cover from ANF interactions to enable Gaussian elimination with negligible overhead.

    - **BASIN** (ours): predicts keys from plaintext–ciphertext pairs, verifies via bitwise SIMON enumeration; ensures correctness, bounded runtime, and stable low-overhead results.

- **Domain-specific optimization**:
    Even limited domain-specific training data can enhance performance via ML-based hyperparameter tuning, with no extra runtime cost.

2. **Dataset scale**

    We clarified the reviewer `YNLq`’s misunderstanding and noted that Appendix A details the experimental settings. Furthermore, we expanded experiments to a size-1000 dataset with consistent results(`YNLq`). Besides, we added: (i) larger-budget hyperparameter tuning(`YNLq,n3Jt`), (ii) NSGA-II optimization(`n3Jt`), and (iii) classical non-ML cryptanalysis solver evaluations(`ytKu`)—further supporting our conclusions.

3. **Documentation** (`cTdE,n3Jt`)

    Following rebuttal rules, we provided more detailed experimental scripts and dataset descriptions in text, and clarified the meaning of “Hyperparameter Optimization.”

---

### Decision · Program_Chairs · 2025-09-18

**Decision:**

Accept (poster)

**Comment:**

A benchmark for evaluating ML-based satisfiability (SAT) solvers is proposed with a focus on cryptographic problem instances. These instances are known to be quite difficult for SAT solvers. Despite being a somewhat niche problem, the reviewers argue that the paper justifies this niche well. Criticisms of the benchmark's code quality were assuaged through the rebuttal process. The experiments receive mixed reviews, and arguably some values like standard deviations, etc., would improve the quality of the paper. Overall, though, the experimental evaluation provides a point of comparison for other works. I recommend its acceptance.